EMBO
Molecular Medicine

# A brain microvasculature endothelial cell-specific viral vector with the potential to treat neurovascular and neurological diseases

Jakob Körbelin[1], Godwin Dogbevia[2], Stefan Michelfelder[1], Dirk A Ridder[2], Agnes Hunger[1], Jan Wenzel[2], Henning Seismann[1], Melanie Lampe[1], Jacqueline Bannach[2], Manolis Pasparakis[3], Jürgen A Kleinschmidt[4], Markus Schwaninger[2],[†] & Martin Trepel[1,5,*,†]

## Abstract

Gene therapy critically relies on vectors that combine high transduction efficiency with a high degree of target specificity and that can be administered through a safe intravenous route. The lack of suitable vectors, especially for gene therapy of brain disorders, represents a major obstacle. Therefore, we applied an *in vivo* screening system of random ligand libraries displayed on adeno-associated viral capsids to select brain-targeted vectors for the treatment of neurovascular diseases. We identified a capsid variant showing an unprecedented degree of specificity and long-lasting transduction efficiency for brain microvasculature endothelial cells as the primary target of selection. A therapeutic vector based on this selected viral capsid was used to markedly attenuate the severe cerebrovascular pathology of mice with incontinentia pigmenti after a single intravenous injection. Furthermore, the versatility of this selection system will make it possible to select ligands for additional *in vivo* targets without requiring previous identification of potential target-specific receptors.

**Keywords** adeno-associated virus; brain microvascular endothelial cells; gene therapy; neurovascular diseases
**Subject Categories** Cardiovascular System; Genetics, Gene Therapy & Genetic Disease; Neuroscience

See also: **S Marchiò et al** (June 2016)

## Introduction

There is a particular medical need for new treatment options for neurovascular and neurological disorders, and gene therapy represents a promising approach to address this need (Nagabhushan Kalburgi *et al*, 2013). However, only few vector systems allow efficient gene transfer to the central nervous system (CNS), and none of them is specific. The low specificity poses a significant risk for side effects. In addition, most of these vectors have to be administered directly into the CNS, which locally restricts gene transfer close to the injection site and thus may not be sufficient for treating disorders with a more widely distributed pathology. Being invasive, such approaches bear the risk of serious complications, including hemorrhages and infections. Instead, targeting brain endothelial cells by intravenous vector injection represents a promising alternative to the direct injection of vectors into the brain parenchyma (Chen *et al*, 2009), as therapeutic gene products may be transported to the abluminal side of the endothelium. Moreover, this strategy may allow for gene therapy of neurovascular disorders.

Vectors based on adeno-associated virus (AAV) are important candidates for clinical application in neurological diseases as they effectively transduce neural cells and have a favorable safety profile (Nagabhushan Kalburgi *et al*, 2013; Weinberg *et al*, 2013). Nonetheless, as with most viral vectors, their natural tropism does not allow specific transduction after systemic administration *in vivo*. Most strategies to target AAVs to defined cell types focus on manipulating the AAV capsid by inserting phage-selected peptides or other ligands (White *et al*, 2004, 2008; Work *et al*, 2004, 2006; Chen *et al*, 2009; Munch *et al*, 2013) or by capsid shuffling (Bowles *et al*, 2003; Maheshri *et al*, 2006; Grimm *et al*, 2008; Koerber *et al*, 2008; Gray *et al*, 2010; Lisowski *et al*, 2014) to abrogate their natural tropism

1  Hubertus Wald Cancer Center, Department of Oncology and Hematology, University Medical Center Hamburg-Eppendorf, Hamburg, Germany
2  Institute of Experimental and Clinical Pharmacology and Toxicology, University of Lübeck, Lübeck, Germany
3  Cologne Excellence Cluster on Cellular Stress Responses in Aging-Associated Diseases (CECAD) and Centre for Molecular Medicine (CMMC), Institute for Genetics, University of Cologne, Cologne, Germany
4  Department of Tumor Virology, German Cancer Research Center, Heidelberg, Germany
5  Department of Hematology and Oncology, Augsburg Medical Center, Augsburg, Germany
*Corresponding author. Tel: +49 821 400 2353; Fax: +49 821 400 3344; E-mail: m.trepel@uke.de
†These authors contributed equally to this work

and redirect them to alternative cellular receptors. Screening of random peptide libraries that are displayed within the structural constraint of the AAV capsid (Muller *et al*, 2003; Perabo *et al*, 2003; Varadi *et al*, 2012) may provide the most naturalistic setting to generate suitable targeted AAV vectors. Using this technology, several vectors with improved transduction properties have been established (Muller *et al*, 2003; Perabo *et al*, 2003; Waterkamp *et al*, 2006; Michelfelder *et al*, 2007; Stiefelhagen *et al*, 2008; Koerber *et al*, 2009; Ying *et al*, 2010; Varadi *et al*, 2012; Deverman *et al*, 2016). However, despite great success in improving transduction efficiency, most of these approaches have failed to significantly enhance vector specificity for predefined target cells after systemic administration *in vivo*. To overcome the shortcomings of *in vitro* selection, we developed the prerequisites for screening random AAV display peptide libraries in mice *in vivo* (Michelfelder *et al*, 2009). In the present study, we applied this approach to select brain-targeted AAV to treat CNS diseases *in vivo*. We generated a vector for the targeted delivery of therapeutic genes to the CNS with a previously unachieved degree of specificity. This vector efficiently transduces the BBB-associated endothelium of the entire murine CNS after a single intravenous injection. As proof of concept, we utilized therapeutic vectors displaying such targeted capsids to markedly attenuate the severe neuropathology in a mouse model of incontinentia pigmenti (IP), a previously untreatable genetic disorder. Since vectors like the one presented here could be selected for virtually any target tissue, our findings have implications not only for the treatment of neurovascular diseases but also for a variety of other non-neurological genetic and nongenetic disorders.

# Results

### *In vivo* screening of a random AAV display peptide library enriches brain-targeted capsids

An AAV2 peptide library displaying random heptapeptide insertions at amino acid position R588 of the capsid protein was intravenously injected into FVB/N mice. After 48 h, the brains were harvested and the DNA of particles that homed to this organ was extracted. The relevant part of the capsid gene was amplified by PCR, re-cloned into a new library, and the selection process was then repeated for four additional rounds. After each round, 10 of the enriched library particles were sequenced (Fig 1A). After the first two rounds of selection, most of the clones displayed peptides sharing the sequence motif $NXX^X_R{}^X_R{}^X_E{}_E$. After the third round, however, a new and clearly distinct sequence motif became apparent, with glycine (G) at position three and tryptophan (W) at position six (XXGXXWX).

To evaluate the targeting potential of the selected capsids, luciferase reporter vectors displaying peptides corresponding to one of the motifs ($NXX^X_R{}^X_R{}^X_E{}_E$ or XXGXXWX) or the wild-type control capsids were intravenously injected into mice. One month after vector administration, CMV promoter-driven luciferase activity was determined from tissue lysates of the brain and control organs (Fig 1B).

The peptide NNVRTSE, which we had also found during *in vitro* screenings on different cell lines (J. Körbelin and M. Trepel, unpublished data) in addition to the *in vivo* screening

reported here, did not mediate enhanced brain transduction (Fig EV1). However, all peptides with the brain-selected variants of the sequence motif XXGXXWX mediated substantially improved transgene expression in the brain compared to wild-type rAAV2 capsids (Fig 1B). The increase in transgene expression in the brain compared to wild-type rAAV2 ranged from eightfold (SDGLAWV) to 65-fold (NRGTEWD), whereas transgene expression in the liver was, if at all detectable, decreased by at least 20-fold (ADGVQWT). The transgene expression profiles of all analyzed library clones, including transduction of potential off-target organs, are shown in detail in Fig EV1.

Adeno-associated virus displaying the NRGTEWD peptide showed the best target specificity with the strongest transgene expression in the brain and minimal or no expression in off-target organs. Therefore, we chose this clone, hence designated AAV-BR1, and further analyzed its transduction profile and investigated its therapeutic potential.

### The brain-selected AAV capsid BR1 (NRGTEWD peptide) efficiently mediates durable and brain-specific gene expression based on specific vector homing *in vivo*

We generated reporter vectors carrying the luciferase gene under the control of the CAG promoter, packaged into either one of three different AAV capsids: AAV2 displaying the peptide NRGTEWD (termed "BR1"), wild-type AAV2, or AAV2 displaying the phage-selected peptide DSPAHPS (termed "PPS") which has been reported to target blood vessels in the brain (Chen *et al*, 2009). Mice injected with AAV-BR1 vector displayed transgene-mediated luminescence of so far unachieved specificity and efficacy in the brain (Fig 2A and Appendix Fig S1). Wild-type rAAV2 capsids, on the other hand, predominantly mediated transgene expression in the liver. The control peptide PPS mediated enhanced overall transgene expression, particularly in the heart and other tissues. The PPS-targeting peptide had been selected in the brain by phage display (Chen *et al*, 2009). Consequently, there was AAV-PPS-mediated gene expression in the brain (see below), but expression was so weak that it could not be detected by *in vivo* imaging. In contrast, the luminescence mediated by the newly selected AAV-BR1 reporter vector was strong enough to be clearly detectable in the brain and the eyes of living animals even at day 264 after vector injection (Fig 2B). The vector's tissue specificity was confirmed by three-dimensional reconstruction and virtual cross sectioning (Fig 2C). Expression of the BR1-mediated transgene was monitored over a prolonged period of time (> 660 days) and proved to be exceptionally stable and target-specific for the entire observation period (Fig 3 and Appendix Fig S2). We also analyzed transgene expression *ex vivo* for greater quantitative accuracy. In tissue lysates of AAV-BR1-transduced mice, luciferase activity in the brain was 1,000-fold stronger than in the liver and 100-fold stronger than in the heart (Fig 4A). Compared to wild-type rAAV2, the same vector dose of AAV-BR1 induced a 650-fold higher transgene expression in the brain. Brain luminescence mediated by AAV-PPS, on the other hand, was 10-fold stronger than in the liver and enhanced 20-fold compared to wild-type rAAV2. However, AAV-PPS-mediated transgene expression was predominantly detected in the heart, where it was approximately 25 times stronger than in the brain. To evaluate whether the brain-specific transgene expression

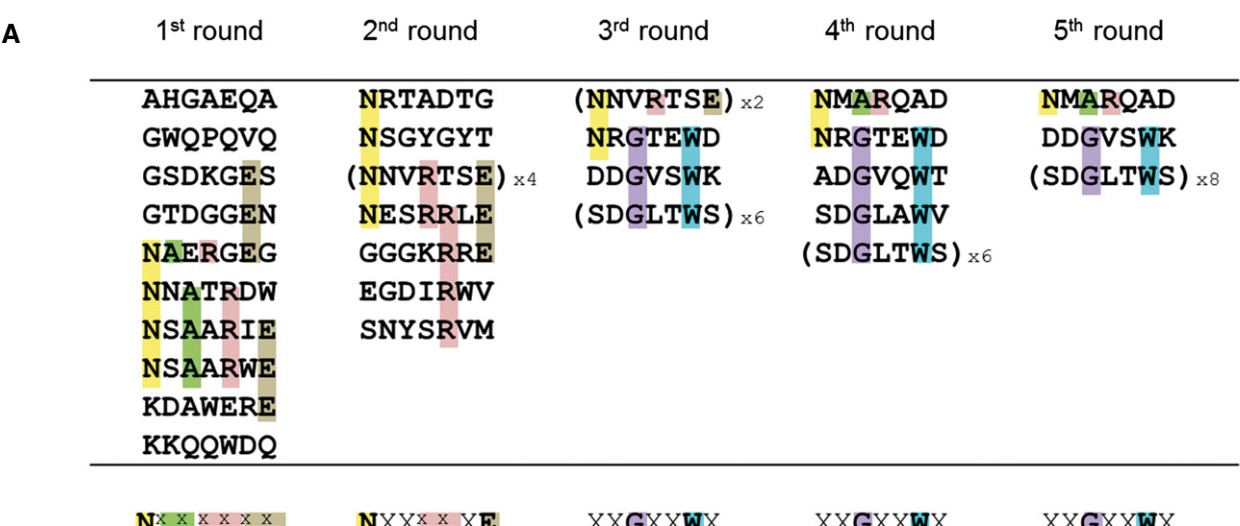

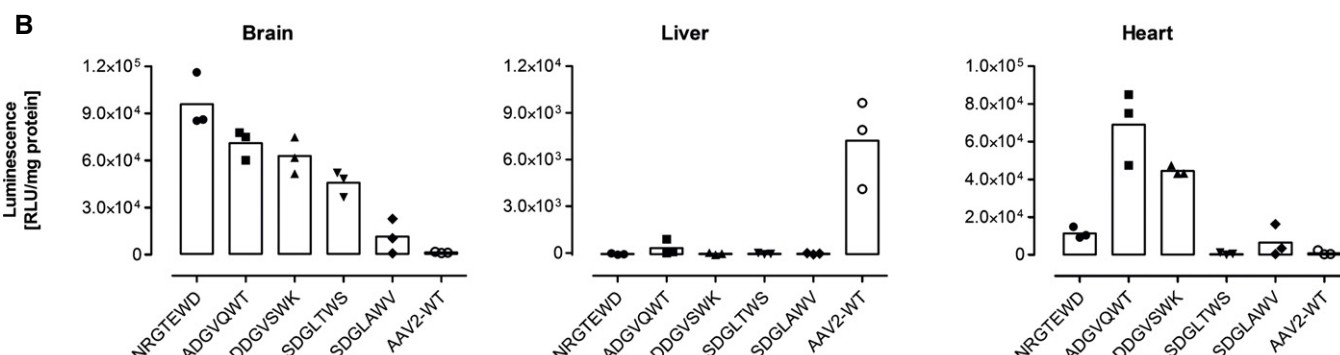

**Figure 1. Capsid variants of a random AAV display peptide library during five rounds of *in vivo* selection enriching for brain-homing library particles.**

A   Randomly chosen library peptide inserts (shown in single-letter code) recovered from the brain. Ten clones were sequenced after each round of selection. Intravenously injected library particles were allowed to circulate for two days in each selection round (*n* = 1 animal/selection round, age 8–12 weeks). Five selection rounds were performed. Amino acids characterizing the emerging motifs are highlighted in different colors. The consensus motif of each selection round is shown at the bottom of each column. Sequences were considered to show a consensus if at least four different clones displayed the same amino acids at the same position or at two adjacent positions.

B   *In vivo* transduction profile of luciferase reporter vectors displaying variants of the enriched library capsid sequence motifs or unmodified wild-type AAV2 control capsid. Four weeks after i.v. injection of $5 \times 10^{10}$ genomic particles/mouse containing a CMV-luciferase reporter gene, luminescence was measured in the brain and control organs. Data are shown as bars (mean) with plotted individual data points (*n* = 3 animals/group, age 8–12 weeks).

was caused by specific peptide-mediated homing, the vector distribution was quantified by qPCR. For all tested vectors, most viral DNA was recovered from spleen, an organ without any detectable transgene expression, indicating nonspecific particle uptake in the reticular endothelial system independently of transduction. When excluding the spleen from analysis, considerable enrichment of AAV-BR1 vector genomes was observed in the brain (Fig 4B). The number of AAV-BR1 vector genomes detected in the brain was over 40 times higher than in the liver and more than 170-fold enriched compared to wild-type rAAV2, whereas an 18-fold decrease in vector genomes was seen in the liver when comparing AAV-BR1 with wild-type rAAV2. While AAV-BR1 showed peptide-mediated vector homing to the brain and de-targeting from the liver, AAV-PPS vector genomes were mainly recovered from the heart and the kidney without being significantly enriched in the brain compared to wild-type rAAV2 (Fig 4B).

**Recombinant vectors displaying the brain-enriched BR1 peptide are highly effective in targeting the blood–brain barrier-associated endothelium of the central nervous system**

To identify the target cells of BR1-mediated transduction, we analyzed brains and control organs of mice that were treated with BR1 vectors carrying GFP under the control of the CAG promoter (Fig 5). Mice treated with AAV-BR1-CAG-eGFP showed intense vector-mediated fluorescence in the microvasculature of the entire brain 2 weeks after vector injection (Fig 5A), without considerable differences in the rate of transduced capillaries between cerebellum, olfactory bulb, striatum, and cerebral cortex (see Appendix Fig S3 for infection rates). In addition, some scattered neuronal transduction was observed, indicating the vector's ability to partially cross the fully developed BBB. Notably, AAV-BR1-mediated transduction of the microvasculature was not restricted to the brain and was also

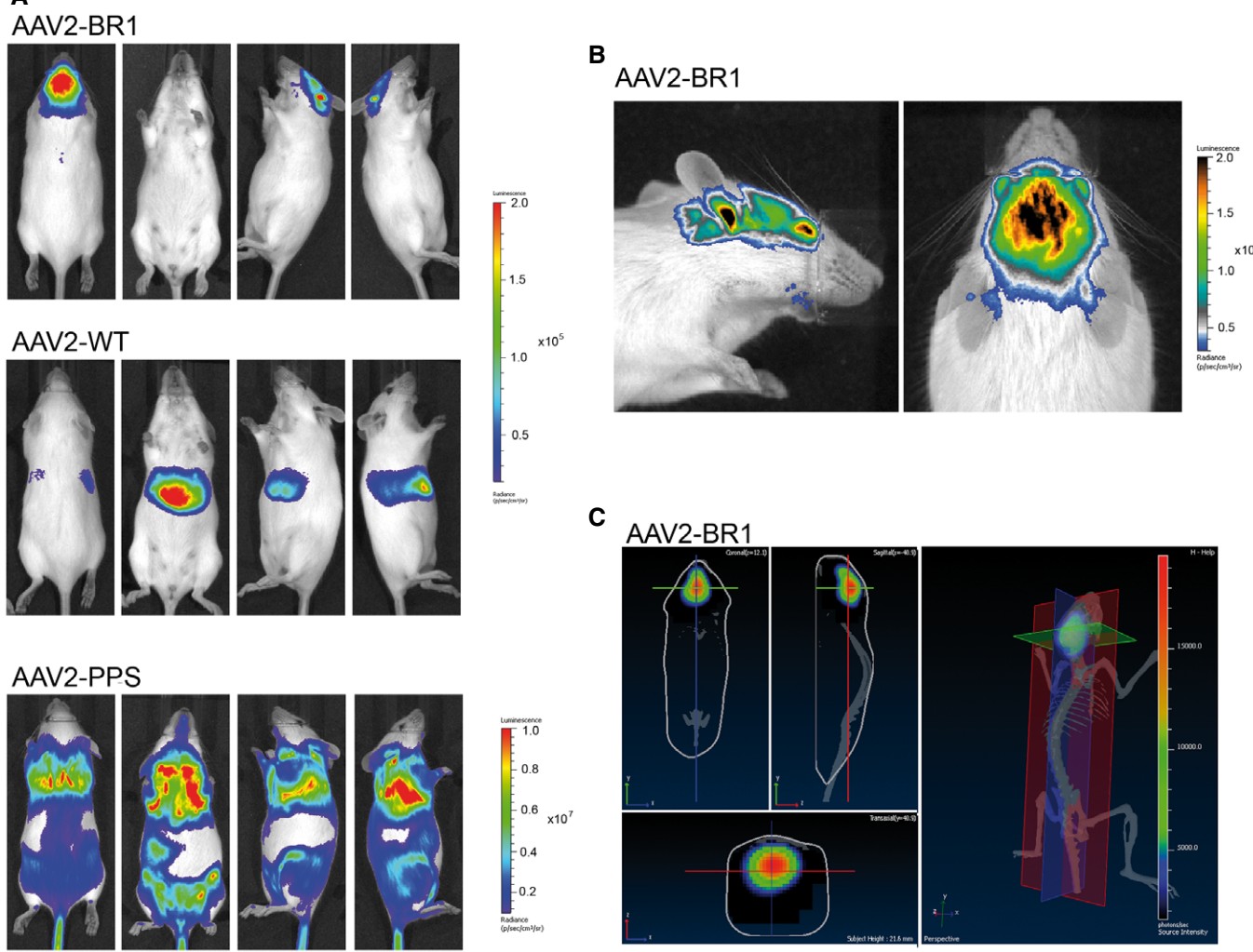

**Figure 2. *In vivo* luminescence imaging of mice after intravenous injection of brain-enriched AAV2 vectors carrying a luciferase reporter gene under the control of the CAG promoter.**

A   Luciferase reporter gene vectors displaying the brain-targeted peptide NRGTEWD ("BR1"), unmodified wild-type AAV2 capsid, or the previously reported brain-targeting peptide DSPAHPS (PPS). Vectors were intravenously injected into mice ($5 \times 10^{10}$ genomic particles/mouse). Panels show representative examples of $n = 5$ animals, age 8–12 weeks per group. Animals were imaged in dorsal (left panel), ventral (second from left panel), and lateral (second from right and right panel) positions, 14 days after vector injection.

B   Close-up imaging of AAV-BR1-treated mice. Mice were imaged in dorsal and lateral positions in a different color scheme allowing detailed visualization of the transduced brain in living animals, even as late as at day 264 after vector injection.

C   Virtual sections: sagittal, coronal, and transaxial (left panels) and three-dimensional reconstruction (right panel) of the luminescence images of a mouse injected with BR1 vector (as in A). Images were obtained by measuring different wavelengths of the emitted light (Living Image software), confirming the brain as exclusive source of luminescence.

seen in the spinal cord, attesting to the vector's targeting properties for the BBB-associated endothelium of the entire CNS (Fig 5A and Appendix Fig S3). In brain sections of mice treated with AAV-BR1-CAG-eGFP, eGFP clearly colocalized with endothelial CD31 staining (Fig 5B) but not with CD13 staining for pericytes (Fig 5C) or aquaporin 4 staining for astrocytic endfeet (Fig 5D). To confirm endothelial cells as the primary target of the vector, we cultured primary cerebral microvascular endothelial cells (PCMECs) from AAV-BR1-treated mice. The majority of the CD31-positive PCMECs showed vector-mediated fluorescence (Fig 5E). The superior transduction of brain microvasculature endothelial cells by AAV-BR1

could also be reconfirmed *in vitro* by infecting murine PCMECs after taking them into culture (Fig EV2). Immortalized human cerebral microvascular endothelial cells (hCMEC/D3) (Weksler *et al*, 2005) were also proven to be susceptible to AAV-BR1. The AAV-PPS control vector showed only marginal infectivity (~10–20%), whereas AAV-BR1 and wild-type AAV2 were equally infective (~40%) for hCMEC/D3 cells in the *in vitro* setting (Fig EV3). To verify the utility of the BR1 vector to modulate the BBB-associated endothelium, Cre reporter mice Ai14 (Madisen *et al*, 2010) were treated with a BR1 vector carrying the Cre recombinase under the control of the CMV promoter. In vector-treated mice, fluorescence was intense

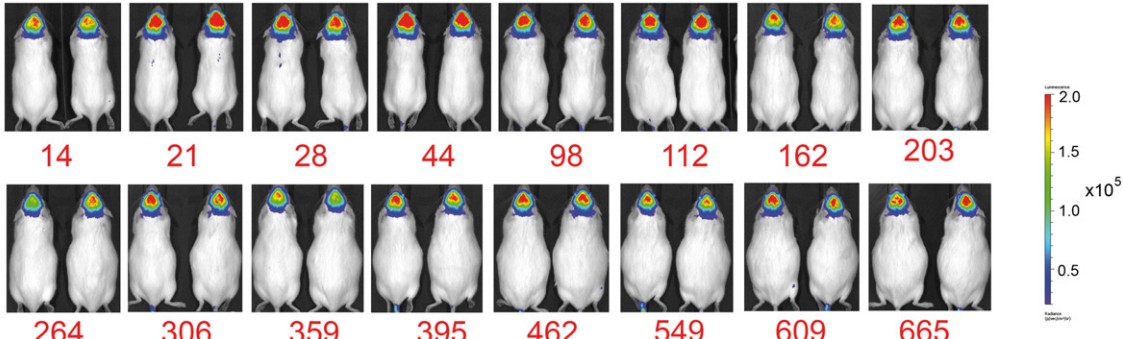

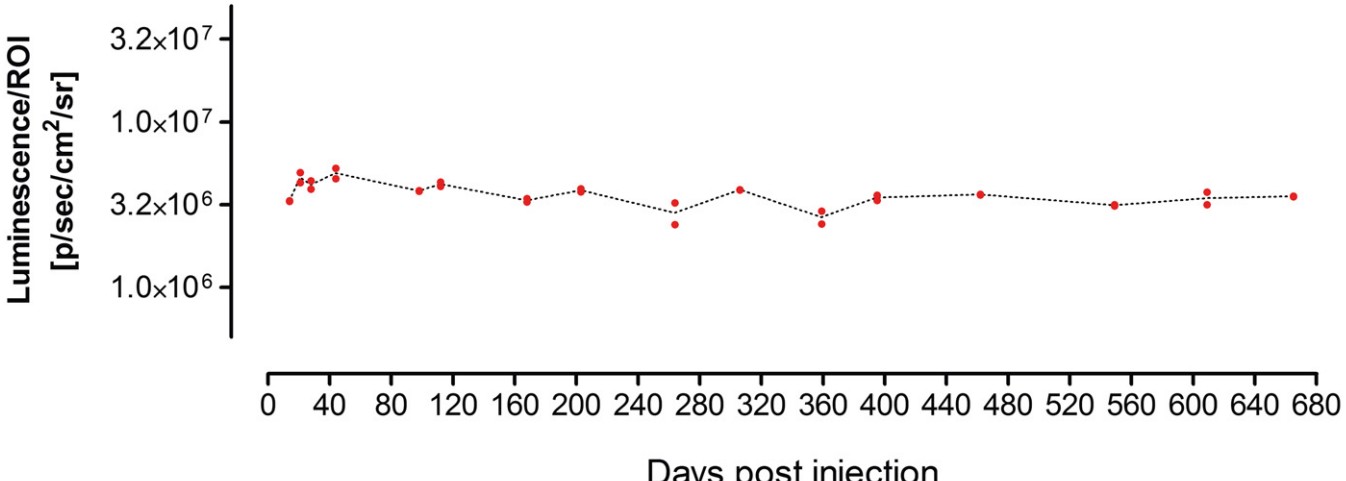

**Figure 3. Long-term transgene expression in the brain mediated by AAV-BR1 vector.**
AAV-BR1 vector harboring the luciferase gene under the control of the CAG promoter was administered intravenously ($5 \times 10^{10}$ genomic particles/mouse, age 8 weeks). Long-term transgene expression was analyzed by *in vivo* bioluminescence imaging at 16 time points during a 665-day period ($n = 2$ animals). Original images of analyzed animals (above) and quantification of luminescence in the brain as the region of interest = ROI (bottom).

in endothelial cells of the entire brain, whereas no such signal could be detected in untreated transgenic mice (Fig 6). Vector-transduced endothelial cells or neurons were barely detectable in the brains of mice treated with either of the control vectors, AAV-PPS or wild-type AAV2 after systemic administration. In the liver and the heart, on the other hand, transduction of hepatocytes and cardiomyocytes was observed for the control vector, but not for the brain-targeted AAV-BR1 vector (Fig EV4). Of note, transduction of brain endothelial cells was also seen when employing alternative injection routes, such as intraperitoneal, intramuscular, or subcutaneous administration of the AAV-BR1 vector (Fig EV5). Nevertheless, intravenous injection proved to be most effective (Fig EV5A) and most specific (Fig EV5B), and can therefore be considered the most favorable route of injection.

**AAV-BR1 vectors can be used to treat vascular disease in the brain in a murine model of incontinentia pigmenti**

To explore the therapeutic potential of the brain endothelial-targeted vector AAV-BR1, we used a mouse model of IP for proof of concept.

This genetic disease is caused by heterozygous, inactivating mutations of the *Nemo* gene on the X chromosome (Smahi *et al*, 2000). IP patients show skin lesions that eventually evolve into changes in dermal pigmentation. However, the major clinical problems are epileptic seizures and other neurological symptoms due to a loss of brain capillaries and a disruption of the BBB as shown in mice deficient of NEMO in brain endothelial cells (Meuwissen & Mancini, 2012; Ridder *et al*, 2015). At postnatal day 8 (P8), numerous empty basement membrane strands without endothelial cells inside, so-called string vessels, are found in brains of $Nemo^{-/+}$ mice (Ridder *et al*, 2015). A representative example of such string vessels is shown in detail in Appendix Fig S4. At P0, the lengths of string vessels had not yet increased in $Nemo^{-/+}$ mice compared to wild-type littermates, opening a therapeutic time window for gene therapy (Fig 7A). When we injected neonatal $Nemo^{-/+}$ mice intravenously at P0 with AAV vectors carrying the Nemo gene under the control of the CAG promoter and displaying the brain-targeted BR1 peptide (AAV-BR1-CAG-NEMO), the lengths of string vessels at P8 were markedly reduced down to the level of wild-type mice (Fig 7A). In addition, the number of apoptotic endothelial cells was

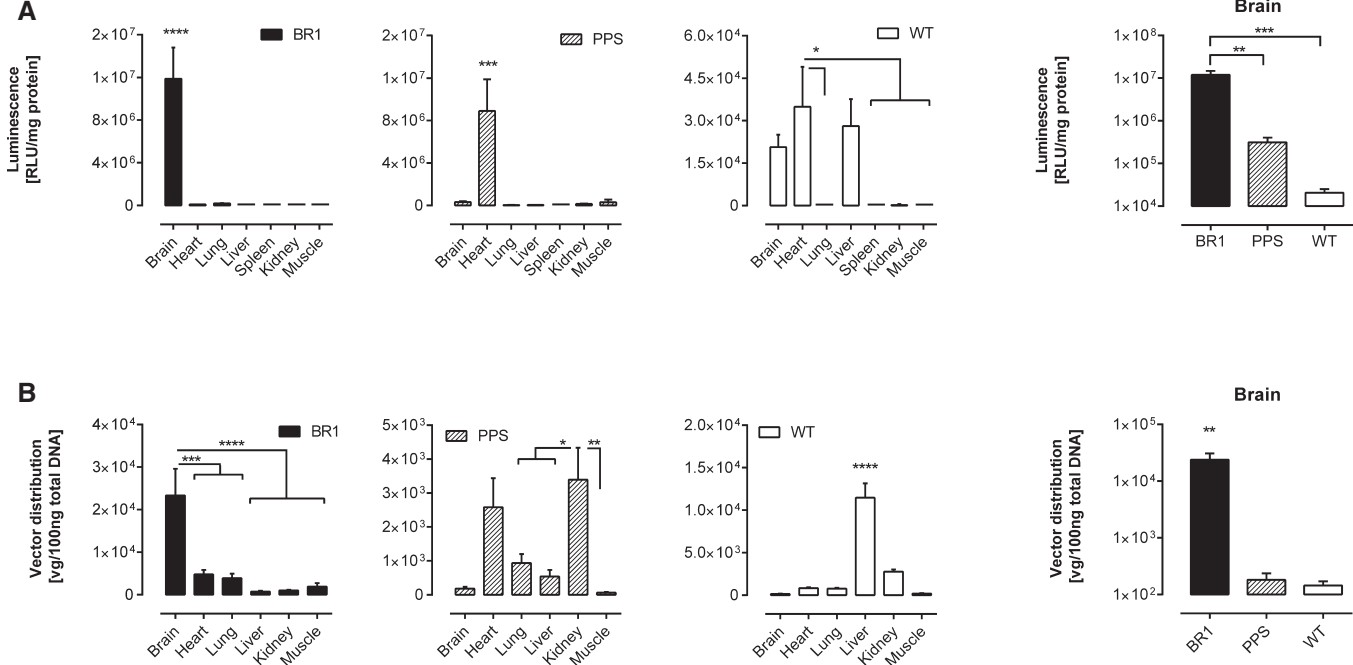

**Figure 4. Luciferase transgene expression and vector biodistribution in tissue lysates.**

Luciferase activity and vector copy numbers were determined in tissue lysates 14 days after vector administration ($5 \times 10^{10}$ genomic particles/mouse, age 8–12 weeks).

A  Vector-mediated luminescence. Transgene expression was measured in AAV-BR1 harboring the luciferase gene under the control of the CAG promoter and control vectors (AAV-PPS and wild-type rAAV2) in brain and off-target control organs (left panel). Comparison of luminescence in the brain mediated by AAV-BR1 and control vectors (right panel). ****$P < 0.0001$ (BR1: brain vs. all), ***$P = 0.0002$ (PPS: heart vs. brain/muscle), ***$P = 0.0001$ (PPS: heart vs. lung/liver/spleen/kidney), *$P = 0.0101$ (WT: heart vs. lung), *$P = 0.0120$ (WT: heart vs. spleen), *$P = 0.0148$ (WT: heart vs. kidney), *$P = 0.0113$ (WT: heart vs. muscle), **$P = 0.0011$ (brain: BR1 vs. PPS), ***$P = 0.0009$ (brain: BR1 vs. WT).

B  Biodistribution of AAV-BR1 and control vectors (AAV-PPS and wild-type rAAV2) in brain and off-target control organs, excluding spleen (left panel). Genome copy numbers of AAV-BR1 and control vectors (AAV-PPS and wild-type rAAV2) in the brain (right panel). ****$P < 0.0001$ (BR1: brain vs. liver/kidney/muscle), ***$P = 0.0006$ (BR1: brain vs. heart), ***$P = 0.0003$ (BR1: brain vs. lung), **$P = 0.0025$ (PPS: kidney vs. muscle), *$P = 0.0378$ (PPS: kidney vs. lung), *$P = 0.0114$ (PPS: kidney vs. liver), ****$P < 0.0001$ (WT: liver vs. all), **$P = 0.0028$ (brain: BR1 vs. PPS/WT).

Data information: Data are shown as mean + SEM, $n = 5$ mice per group. Data were analyzed by one-way ANOVA, followed by Turkey's multiple comparison test. The mean of each column was compared to the mean of the column with the highest value.

significantly lower in AAV-BR1-CAG-NEMO-treated than in control vector-treated animals ($P = 0.0201$; Fig 7B), demonstrating that transducing *Nemo* rescues the survival of brain endothelial cells in $Nemo^{-/+}$ mice. Moreover, AAV-BR1-CAG-NEMO administration to $Nemo^{-/+}$ mice normalized the extravasation of albumin into brain tissue, indicating that targeted delivery of the Nemo gene into brain vessels is able to maintain an intact BBB in $Nemo^{-/+}$ mice (Fig 7C). The remarkable improvement of neuropathology was accompanied by a higher body weight of $Nemo^{-/+}$ mice after administering AAV-BR1-CAG-NEMO (Fig 7D).

In contrast to humans, the skin phenotype is associated with a high mortality of the $Nemo^{-/+}$ genotype in mice between P8 and P12 (Nenci *et al*, 2006). In order to study the functional consequences of gene therapy in adult animals despite the high mortality of $Nemo^{-/+}$ mice (Schmidt-Supprian *et al*, 2000), we adopted a conditional approach to delete Nemo specifically in brain endothelial cells ($Nemo^{beKO}$). As reported previously, the cerebrovascular pathology of $Nemo^{beKO}$ mice mimicked the situation in $Nemo^{-/+}$ animals (Ridder *et al*, 2015). Intravenous injection of AAV-BR1-CAG-NEMO into adult $Nemo^{beKO}$ mice (followed by 5 days of tamoxifen treatment, starting 1 week after vector administration) reduced

the number of string vessels (Fig 8A and B), normalized the total vessel lengths (Fig 8C), and reversed the extravasation of albumin and immunoglobulin into brain tissue (Fig 8D and E). Importantly, infection of brain endothelial cells had no effect on the tightness of the BBB in wild-type animals (Fig 8F), suggesting a unique and safe means to tighten the BBB, protect cerebral vessels, and supply proteins to the CNS.

# Discussion

The future of gene therapy for neurological diseases will depend on the development of safe, efficient, and target-specific vectors. Administration of such vectors should be technically easy, and the vectors should be able to reach the entire CNS through the circulation and not just circumscribed areas amenable to local application. For a long time, the ability to cross the BBB was thought to be a prerequisite for CNS-directed gene therapy vectors. However, transduction of the CNS-associated endothelium by targeted vectors seems to be sufficient to induce therapeutic effects at least in some CNS disorders, due to the close proximity of neurons and

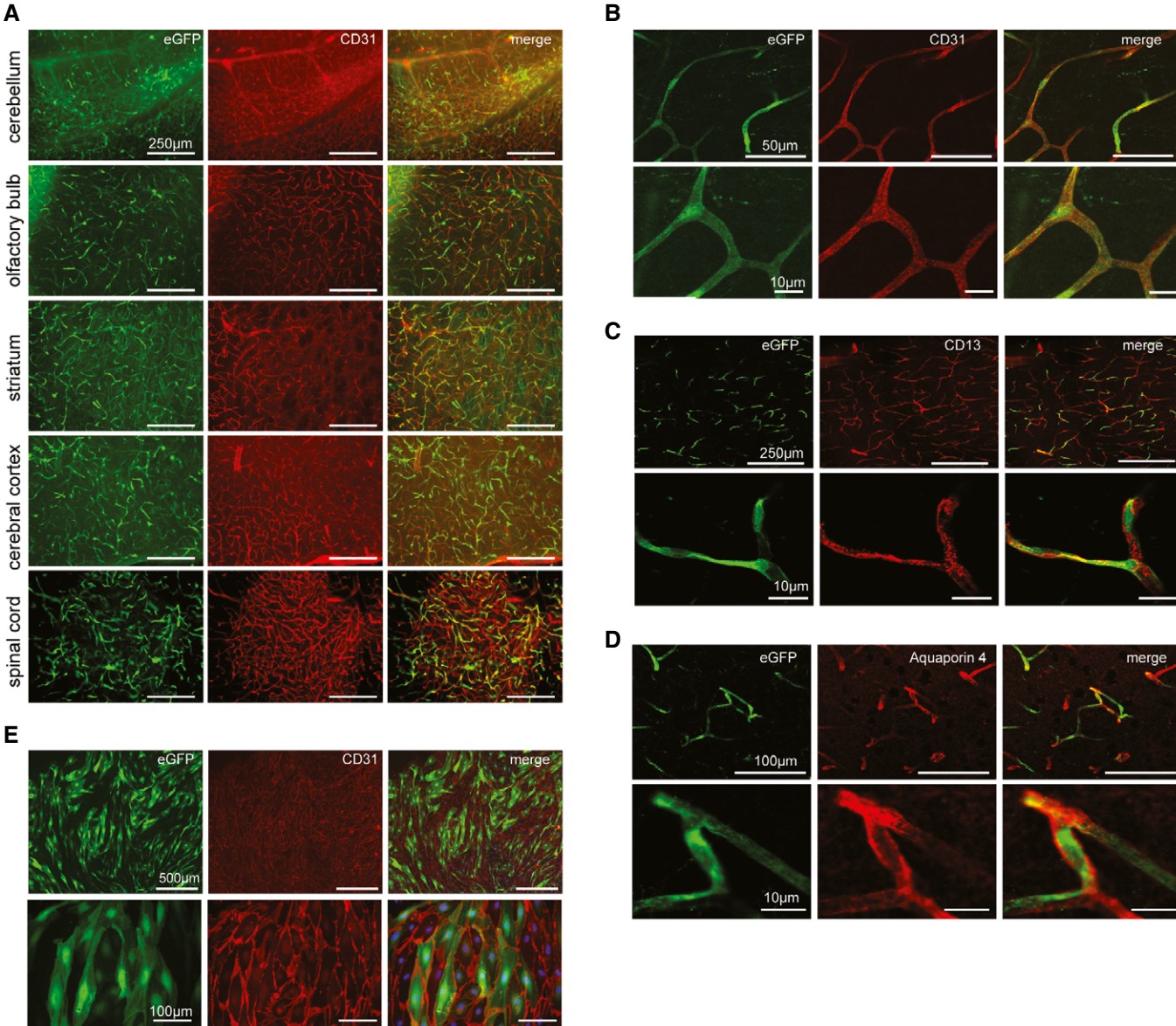

**Figure 5. AAV-BR1-mediated transduction of brain endothelial cells *in vivo*.**

C57BL/6 mice (age 8 weeks) were injected with AAV-BR1 harboring an eGFP reporter gene under the control of the CAG promoter. Images show representative examples of *n* = 6 mice.

A   Representative images from cerebellum, olfactory bulb, striatum, cerebral cortex, and the spinal cord, 14 days after vector injection. BR1-eGFP-transduced cells (green) were positive for the endothelial marker CD31 (red). Scale bars represent 250 μm.

B   Higher magnification confocal images. The endothelial marker CD31 (red) colocalizes with vector-mediated eGFP expression (green). Scale bars represent 50 μm (upper panel) or 10 μm (lower panel).

C   CD13 staining of pericytes. The vector-mediated eGFP expression pattern (green) does not colocalize with CD13 (red). Scale bars represent 250 μm (upper panel) or 10 μm (lower panel).

D   Aquaporin 4 staining of astrocytic endfeet. The vector-mediated eGFP expression (green) does not colocalize with aquaporin 4 (red). Scale bars represent 100 μm (upper panel) or 10 μm (lower panel).

E   Expression of eGFP in primary brain endothelial cells prepared from C57BL/6 mice, 14 days after injection with AAV-BR1-eGFP. Scale bars represent 500 μm (upper panel) or 100 μm (lower panel).

endothelial cells. In mouse models of two lysosomal storage diseases, CNS pathology could be ameliorated by overexpressing the relevant enzymes in brain endothelial cells (Chen *et al*, 2009). Thus, it seems likely that endothelial expression of proteins with relevance for other neurological disorders (e.g., for one of the numerous other lysosomal storage diseases, Canavan disease, ataxia telangiectasia) and their subsequent transport into the brain parenchyma might be a promising treatment option. Despite such ground-breaking discoveries, currently available vectors lack the degree of tissue specificity and efficacy that is necessary to mediate strong and long-lasting therapeutic effects while minimizing side effects, not only for CNS disease; especially, the discussion about

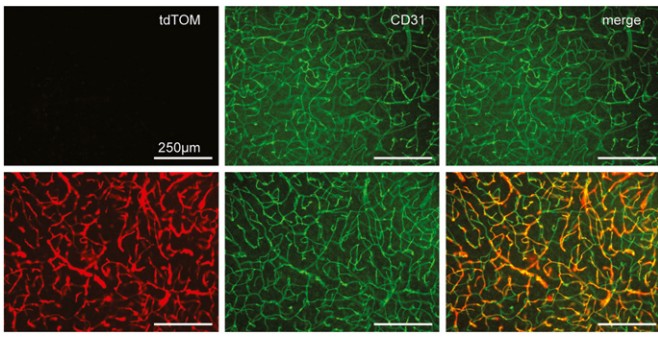

**Figure 6.    AAV-BR1-iCre-mediated gene recombination in Ai14 Cre reporter mice.**
Two weeks after vector injection (1.8 × 10¹¹ genomic particles/animal) into 16-week-old mice, Cre-mediated gene recombination driven by the CAG promoter was observed mainly in brain endothelial cells (lower panel, red). No recombination was observed in control animals without BR1-iCre-virus injection (upper panel). CD31 (green) was used as a marker for brain endothelial cells. Panels show representative examples of *n* = 3 animals. Scale bars represent 250 μm.

hepatocellular carcinomas (HCC) that are being induced by recombinant AAV vectors in animal models (Donsante *et al*, 2001, 2007; Bell *et al*, 2006; Montini, 2011; Rosas *et al*, 2012; Valdmanis *et al*, 2012) and the recent reports on wild-type AAV2-induced HCC in humans (Nault *et al*, 2015; Russell & Grompe, 2015) emphasize the need for more target-specific AAV vectors for intravenous injections.

Several AAV serotypes allow neuronal transduction after being locally administered (McCown *et al*, 1996; During *et al*, 1998; Mandel *et al*, 1998; Davidson *et al*, 2000; Cearley & Wolfe, 2006; de Backer *et al*, 2010; Markakis *et al*, 2010; Shen *et al*, 2013), and some of them (e.g., AAV9, AAVrh.8, and AAVrh.10) seem to be able to cross the mature BBB after systemic administration in adult animals (Foust *et al*, 2009; Dayton *et al*, 2012; Yang *et al*, 2014). However, very high vector doses (up to 2 × 10¹⁴ vg/kg) are needed to efficiently transduce the CNS, and the brain endothelial cells—being a very important therapeutic target as key player of the BBB—are not transduced. In addition, none of the natural AAV serotypes is CNS-specific. This is especially true for serotypes such as AAV9,

which is commonly used to target the brain, albeit showing much stronger tropism for different off-target tissues, including heart, liver, and skeletal muscles (Zincarelli *et al*, 2008). Vectors with considerably improved transduction efficiency for various target tissues have been generated by conjugating receptor-targeted antibodies or other ligands to the AAV capsid (Bartlett *et al*, 1999; Ponnazhagan *et al*, 2002; Ried *et al*, 2002; Munch *et al*, 2013). Other attempts aimed at changing the natural AAV tropism by incorporating small, previously selected (e.g., by phage display) targeting peptides directly into the receptor-binding region (White *et al*, 2004, 2008; Work *et al*, 2004, 2006; Chen *et al*, 2009). However, such approaches rely on the (limited) availability of specific targeting ligands and their (even more limited) compatibility with vector transduction beyond receptor binding. Frequently, the targeting properties of preselected peptides change when transferring them into the protein context of a viral capsid, rendering this approach to developing vectors unpredictable. The screening of random AAV display peptide libraries is the only systematic approach by which peptides can be selected directly within the structural constraints of the assembled AAV capsid (Muller *et al*, 2003; Perabo *et al*, 2003; Waterkamp *et al*, 2006; Varadi *et al*, 2012). Although the screening of such libraries has yielded numerous vectors well suited for *in vitro* experiments (Muller *et al*, 2003; Perabo *et al*, 2003; Waterkamp *et al*, 2006; Michelfelder *et al*, 2007; Varadi *et al*, 2012), no efficient and tissue-specific vectors for systemic *in vivo* administration in the brain have been developed so far. Also a new AAV9 vector which has been selected recently by a Cre-dependent strategy lacks CNS specificity (Deverman *et al*, 2016). Based on initial studies demonstrating the general feasibility of screening AAV display peptide libraries *in vivo* (Michelfelder *et al*, 2009; Ying *et al*, 2010), we have now applied this screening technology to generate a vector with high degree of specificity and a unique transduction efficacy for the vascular endothelium of the brain, the spinal cord, and presumably also of the inner retina of mice. The additional detection of sporadically transduced neurons in different areas of the brain indicates that AAV-BR1 might to some degree be able to traffic through the endothelial layer via transcytosis and to cross the BBB. Although it is still unknown which receptor is employed by AAV-BR1 for infecting brain endothelial cells, it is likely that the type I transmembrane protein KIAA0319L might be employed for cell entry by AAV-BR1, as this protein has very recently been shown to

**Figure 7.    Therapeutic use of AAV-BR1: normalizing endothelial cell survival and blood–brain barrier permeability in neonatal incontinentia pigmenti mice (*Nemo*⁻/⁺) with AAV-BR1-NEMO.** ▶
BR1-mediated expression of NEMO or eGFP was driven by the CAG promoter.

A    Immunostaining of cerebral microvessels. Treatment with AAV-BR1-NEMO normalized string vessels (white arrows, highlighted in white square inset) in *Nemo*⁻/⁺ mice as compared to *Nemo*⁻/⁺ mice treated with the AAV-BR1-eGFP control vector at postnatal day 8 (P8). The upper left panel shows the staining in untreated wild-type control mice. String vessels were identified as capillaries that have lost CD31-positive (green) endothelial cells but stain for the basement membrane protein collagen IV (red). Scale bars represent 200 μm. The lower right panel summarizes quantitative analysis of string vessels in *Nemo*⁻/⁺ and control mice (*Nemo*ᶠˡ or wild-type) at P0 (*n* = 3 animals/group) and at P8 (*n* = 6 animals/group, *P* = 0.0125). String vessels in the cerebral cortex were quantified as percentage of total vessel lengths.

B    Quantification of active caspase-3-positive endothelial cells at P8 (*n* = 5 animals/group, *P* = 0.0201).

C    Albumin in brain tissue as indicator for BBB leakage. In *Nemo*⁻/⁺ mice treated with AAV-BR1-NEMO, less albumin was found in brain tissue, indicating less BBB leakage. Representative Western blot with albumin from P8 *Nemo*⁻/⁺ mice treated with AAV-BR1-NEMO or AAV-BR1-eGFP control vector, respectively, as well as untreated wild-type (WT) control mice. Right panel: quantitative analysis of the Western blots (*n* = 4 animals/group, *P* = 0.0283).

D    Body weight of vector-treated mice. WT control or *Nemo*⁻/⁺ mice at P8 treated with AAV-BR1-NEMO or AAV-BR1-eGFP (*n* = 18 animals/group, *P* = 0.0038).

Data information: Data are shown as mean + SEM or as plotted individual points with bars representing the mean. Differences between vector-treated *Nemo*⁻/⁺ mice were analyzed by unpaired *t*-test.
Source data are available online for this figure.

be essential for cellular uptake of a variety of AAV serotypes (Pillay *et al*, 2016). In addition, the carbohydrate structure on the endothelial cell surface is likely to be essential for specific targeting since all known primary attachment receptors of AAV are carbohydrates, namely glycans (Mietzsch *et al*, 2014).

In terms of efficient and specific transduction of the brain, the AAV vector presented here strongly outperforms previously

described natural AAV serotypes (Foust *et al*, 2009; Dayton *et al*, 2012; Yang *et al*, 2014) and AAV vectors generated by inserting phage-selected peptides. This demonstrates the advantage of the AAV library system, which selects targeting ligands in the structural and functional context needed for subsequent therapeutic application. In addition, the AAV-BR1-mediated transgene expression in the brain was unexpectedly persistent. Considering recombinant

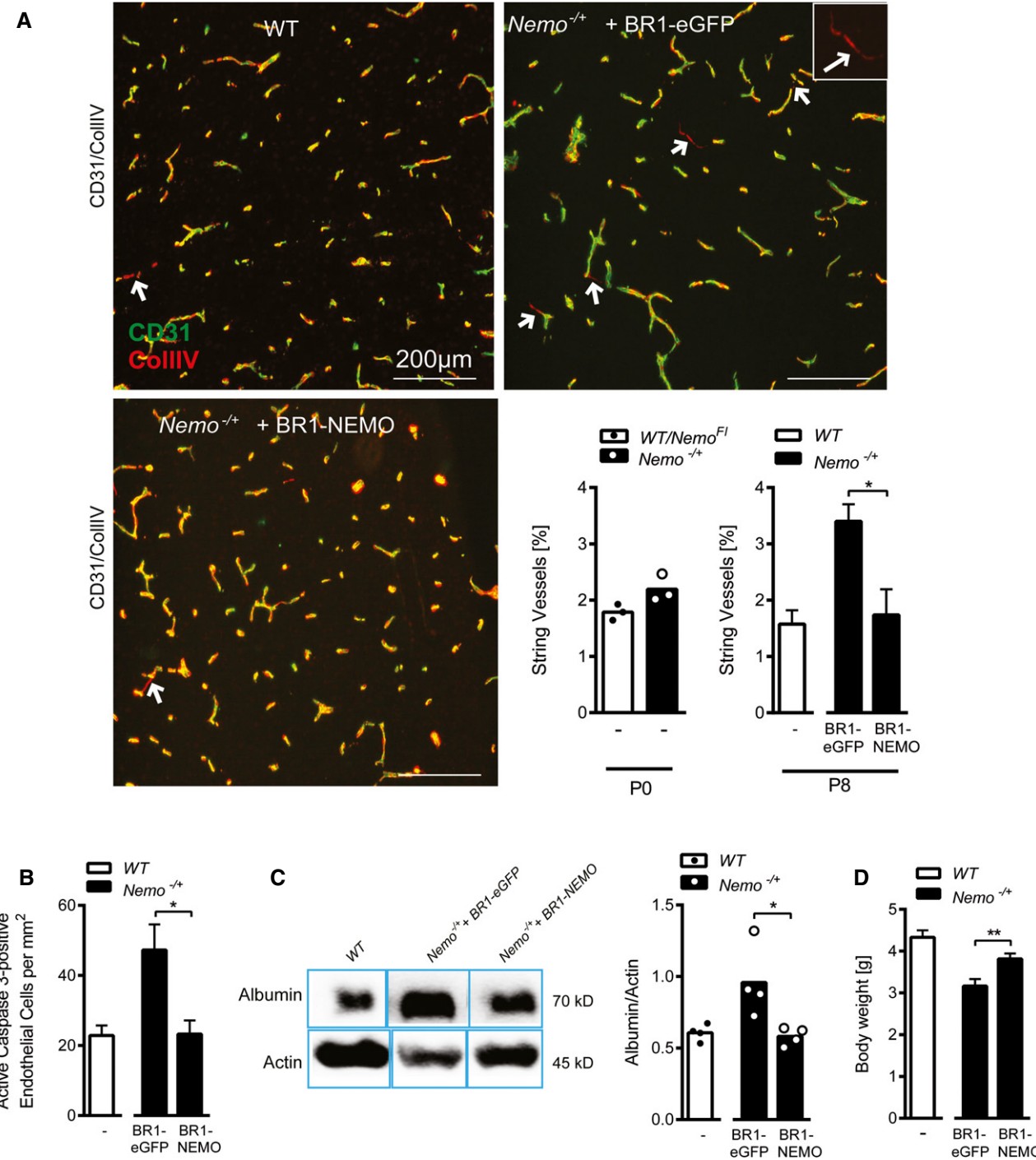

**Figure 7.**

**Figure 8.**

◀

**Figure 8.  Normalizing endothelial cell survival and blood–brain barrier permeability by intravenous injection of AAV-BR1-NEMO in a conditional murine incontinentia pigmenti model (*Nemo^beKO* mice).**

BR1-mediated expression of NEMO or eGFP was driven by the CAG promoter. All animals were at the age of 8–12 weeks.

A   Representative immunostainings of cerebral microvessels. String vessels (white arrows, highlighted in white square inset) were significantly reduced in *Nemo^beKO* mice treated with AAV-BR1-NEMO compared to *Nemo^beKO* mice treated with AAV-BR1-eGFP control vector. *Nemo^Fl* mice served as control animals. Scale bars represent 200 μm.

B   Quantification of string vessel lengths in the cerebral cortex as percentage of total vessel lengths. *Nemo^beKO* and *Nemo^Fl* mice were treated with AAV-BR1-NEMO or AAV-BR1-eGFP control vector (n = 9 *Nemo^Fl* animals + BR1-eGFP, 10 *Nemo^beKO* animals + BR1-eGFP, 9 *Nemo^Fl* animals + BR1-NEMO, and 13 *Nemo^beKO* animals + BR1-NEMO), ****$P < 0.0001$.

C   Total vessel length measured as total CD31-positive vessels. Vessels were restored in *Nemo^beKO* mice treated with AAV-BR1-NEMO compared to the AAV-BR1-eGFP injected mice. Nemo^Fl mice served as a control (n = 9 *Nemo^Fl* animals + BR1-eGFP, 10 *Nemo^beKO* animals + BR1-eGFP, 9 *Nemo^Fl* animals + BR1-NEMO, and 13 *Nemo^beKO* animals + BR1-NEMO), ***$P = 0.0007$, **$P = 0.0038$ (NemoKO:eGFP vs. NemoFl:NEMO), **$P = 0.0014$ (NemoKO:NEMO vs. NemoKO:eGFP).

D   IgG and albumin Western blots of brain lysates. Less leakage in the BBB was detected in *Nemo^beKO* mice treated with AAV-BR1-NEMO than in *Nemo^beKO* mice injected with AAV-BR1-eGFP (the right panel indicates the quantified gel intensity under the various treatment conditions; n = 5 animals per group). *Nemo^Fl* mice served as controls, ****$P < 0.0001$.

E   Quantitative immunoglobulin staining of coronal brain sections of *Nemo^Fl* and *Nemo^beKO* mice. Ig extravasation was significantly reduced in AAV-BR1-NEMO-treated mice (n = 9 *Nemo^Fl* animals + BR1-eGFP, 10 *Nemo^beKO* animals + BR1-eGFP, 9 *Nemo^Fl* animals + BR1-NEMO, and 13 *Nemo^beKO* animals + BR1-NEMO), ****$P < 0.0001$.

F   Effect of AAV-BR1 vector on BBB permeability. No vector (left) or empty AAV-BR1 vector (right) was injected i.v. to wild-type mice and BBB permeability was assessed by extravasation of the fluorescent tracer sodium fluorescein (n = 7 animals per group). No significant difference was detected (n.s.).

Data information: Data are shown as mean + SEM. Data were analyzed by two-way ANOVA followed by Bonferroni's post-test (A–E) or by Student's *t*-test (F).
Source data are available online for this figure.

AAVs as mostly nonintegrating vectors, the observed stability of expression may most likely be explained by the slow turnover rate of the transduced brain endothelial cells.

Overall, the *in vivo* tropism of AAV-BR1 could be confirmed *in vitro* in primary murine and immortalized human brain endothelial cells. The fact that AAV-BR1 and wild-type AAV2 were equally effective in infecting the human hCMEC/D3 cell line (Weksler *et al*, 2005) might be explained by potential inter-species differences (Lisowski *et al*, 2014) or, not less likely, by differences between the situation *in vivo* and *in vitro*. Endothelial cells are prone to dedifferentiate and to change their expression profile when being taken into culture (Liaw & Schwartz, 1993; Staton *et al*, 2009). Moreover, the blood flow *in vivo* and the constant incubation in cell culture medium *in vitro* lead to different exposure times for particles to attach to the receptor. Thus, the prognostic value of these *in vitro* experiments for the potential benefit of AAV-BR1 in a clinical setting is limited.

As a crucial component of blood vessels in the brain and BBB, brain endothelial cells are involved in numerous neurological diseases, including primary vascular diseases such as stroke or vascular-mediated diseases such as multiple sclerosis, and even in tumor growth and epilepsy. Transducing these cells with the AAV-BR1 vector may represent a convenient and safe way to improve cerebral perfusion, to modulate the BBB, or to improve drugability of CNS targets beyond the BBB via a peripheral route. Our data in the IP model provide proof of principle for this strategy. Transducing *Nemo* into brain endothelial cells *in vivo* ameliorated CNS involvement that is the leading manifestation of this disease in humans (Meuwissen & Mancini, 2012). So far, no specific treatment of IP is available. Importantly, by sparing peripheral cells—including peripheral, non-CNS endothelial cells—gene therapy with the AAV-BR1 vector obviates potential adverse effects that have been attributed to NEMO, such as promoting atherosclerosis (Gareus *et al*, 2008). Notably, vectors such as the one presented here not only could be employed to treat IP but can also be applied to treat a broad range of other severe neurological diseases. Since the screening of random AAV display peptide libraries is not limited to organs such as the brain, the strategy presented in our study might help to improve vector development for a large spectrum of diseases.

## Materials and Methods

### Preparation of the random AAV2 display peptide library

A random AAV2 display library displaying heptapeptide insertions at amino acid position R588 (VP1 numbering) with a diversity of $1 \times 10^8$ unique clones (calculated at plasmid level) was produced using a three-step protocol as described previously (Muller *et al*, 2003; Waterkamp *et al*, 2006). In short, a degenerate oligonucleotide encoding seven random amino acids (encoded by NNK to avoid two out of three stop codons and to limit the number of degenerate codons) was synthesized commercially (Metabion) as follows: 5′–CAGTCGGCCAGAGAGGC(NNK)₇GCCCAGGCGGCTGACGAG–3′. Second-strand synthesis was performed using the Sequenase enzyme (Affymetrix) and the primer 5′–CTCGTCAGCCGCCTGG–3′. The double-stranded oligonucleotide insert was cleaved with *Bgl*I, purified with the QIAquick Nucleotide Removal Kit (Qiagen), and ligated into the *Sfi*I-digested pMT187-0-3 library plasmid (Muller *et al*, 2003) at nucleotide position 3967 of the AAV genome. Electrocompetent DH5α bacteria were transformed with the library plasmids using the Gene Pulser Electroporation System (Bio-Rad). The diversity of the plasmid library was determined by the number of clones growing from a representative aliquot of the transformed bacteria on LB agar containing 150 mg/ml ampicillin. Library plasmids were harvested from transformed bacteria and purified using the NucleoBond PC100 plasmid preparation kit (Macherey-Nagel). The AAV library genomes were packaged into chimeric capsids (AAV transfer shuttles) as a hybrid of wild-type and library AAV capsids by transfecting $2 \times 10^8$ 293T cells in ten 15-cm cell culture dishes at a 1:1:2 ratio of the plasmid pVP3 cm (Waterkamp *et al*, 2006) (containing the wild-type cap gene with modified codon usage, without inverted terminal repeats), the library plasmids (Muller *et al*, 2003) (containing the cap gene with peptide insertion, with inverted terminal repeats as packaging signal), and the pXX6 adenoviral helper plasmid (Xiao *et al*, 1998). The resulting AAV library transfer shuttles were used to infect $2 \times 10^8$ 293T cells in ten 15-cm cell culture dishes at a multiplicity of infection (MOI) of 0.5 replicative units per cell. Cells were superinfected with Ad5 at an MOI of 5 plaque-forming units (pfu)/cell. The final random peptide

AAV display library was harvested approximately 72 h after infection from the supernatant of the lysed cells. The supernatant was concentrated using VivaSpin columns (Viva Science), and library particles were purified by iodixanol density-gradient ultracentrifugation (Zolotukhin *et al*, 1999). Thus, a discontinuous iodixanol gradient was prepared by subsequently underlying the harvested library particles with 15, 25, 40, and 54% iodixanol solutions, followed by ultracentrifugation at 230,000 *g* for 70 min. The purified library particles were aspirated from the layer containing 40% iodixanol and dialyzed against HBSS. The virus titer was determined from 1:10,000 diluted samples at the genomic level by real-time PCR (see below) using the *cap*-specific primers 5′–GCAGTATGGTTCTG TATCTACCAACC–3′ and 5′–GCCTGGAAGAACGCCTTGTGTG–3′.

### Determination of vector copy numbers by qPCR

To determine vector copy numbers, we used the SYBR Green I-based FastStart Essential DNA Green Master (Roche) with the LightCycler Nano System (Roche) or the Platinum SYBR Green qPCR Supermix (Invitrogen) with the ABI Prism 7000 Sequence Detection system (Applied Biosystems). An initial denaturation of the probes (95°C, 10 min) was followed by 45 cycles of amplification (95°C/67°C/72°C; 30 s each; rampage 5°C/s) and a final melting curve analysis (60–97°C with 0.1°C/s). For each vector, corresponding plasmid DNA was used to generate a standard curve. Primers used for titration were either *cap*-specific (AAV library) or promoter-specific (recombinant AAV vectors).

### *In vivo* screening of the random AAV2 display peptide library

For the *in vivo* selection, $1 \times 10^{11}$ genomic library particles were injected into the tail vein of FVB mice. Two days later, mice were sacrificed and the complete brain as organ of interest was removed. Total tissue DNA was extracted using the DNeasy Tissue Kit (Qiagen). The random oligonucleotide insertions from the enriched AAV library particles were amplified by nested PCR using the primers 5′–ATGGCAAGCCACAAGGACGATG–3′ and 5′–CGTGGAGTACTGTG TGATGAAG–3′ for the first PCR, and 5′–GGTTCTCATCTTTGGGAA GCAAG–3′ as well as 5′–TGATGAGAATCTGTGGAGGAG–3′ for the second PCR. Approximately 20 PCRs with 1 μg template DNA each were set up for the two PCR rounds. The PCR-amplified oligonucleotides were used to produce preselected libraries for subsequent rounds of selection. Preselected libraries were produced like the primary library (as described above). Five rounds of selection were performed in *n* = 1 animal each. After each round of selection, ten clones were sequenced. Selected library clones were produced as recombinant AAV vectors for further analysis.

### Plasmid constructs for Nemo gene transfer experiments

The mouse IKKγ/*Nemo* gene Ikbkg (accession number NM_001136067) was amplified by PCR from mouse endothelial cDNA with the forward primer BstB1: 5′–CCGATTCGAATTCA CCATGTATATCAGGTAC–3′ and the reverse primer Xho1: 5′–TGCCCTCGAGCTCTATGCACTCCATGACATG–3′. The 1,306-bp *Nemo* fragment was cloned into the vector pAAV-CAG-BMP2-2A-eGFP (Heinonen *et al*, 2014) by removing BMP2 with *Bst*BI and *Xho*I and replacing it with Nemo, generating the plasmid

pAAV-CAG-NEMO-2A-eGFP. The plasmid pAAV-CMV-iCre-2A-eGFP was generated as previously described (Heinonen *et al*, 2014).

### Vector production and quantification

Recombinant AAV vectors were produced by triple transfection of HEK293T cells. Cells were grown at 37°C, 5% $CO_2$ in DMEM (Invitrogen) supplemented with 1% penicillin/streptomycin (Invitrogen) and 10% fetal calf serum (Biochrom). HEK293T cells were transfected with plasmid DNA using linear polyethylenimine (Polysciences) or by the calcium phosphate method (Grimm *et al*, 2003; Zhu *et al*, 2007). Three days after transfection, cells were harvested and then lysed by repeated freeze–thaw cycles in PBS-MK, and the vectors were then purified by iodixanol density-gradient ultracentrifugation (see above) or via affinity column purification using Sepharose columns (HiTrap™, AVB Sepharose™; GE Healthcare). For transfections, we used pXX6 or p179 as adenoviral helper plasmid (Xiao *et al*, 1998), the luciferase reporter plasmids pAAV-CMV-LUC (containing inverted terminal repeats of AAV2, the CMV promoter, the firefly luciferase gene, and the SV40 Poly-A signal), pAAV-CAG-LUC (containing inverted terminal repeats of AAV2, the CAG promoter, the firefly luciferase gene, and the SV40 Poly-A signal), the GFP reporter plasmid pAAV-CAG-eGFP (containing inverted terminal repeats of AAV2, the CAG promoter, the eGFP gene, and the SV40 Poly-A signal), the plasmid pAAV-CAG-NEMO-2A-eGFP or pAAV-CMV-iCre-2A-eGFP (see above), and a plasmid encoding the modified AAV capsid of interest. Plasmids encoding the AAV capsid mutants and wild-type controls were modified pXX2-187 (Michelfelder *et al*, 2007) and pXX2 (Xiao *et al*, 1998; Chen *et al*, 2009).

To quantify vector stocks, genomic titers were determined by quantitative real-time PCR (see above) using the CAG-specific primers 5′–GGACTCTGCACCATAACACAC–3′ and 5′–GTAGGAAAG TCCCATAAGGTCA–3′ for the plasmids pAAV-CAG-LUC and pAAV-CAG-eGFP and the WRPE-specific primers 5′-TGCCCGCTGCTGGAC-3′ and 5′-CCGACAACACCACGGAATTG-3′ for the plasmids pAAV-CAG-NEMO-2A-eGFP and pAAV-CMV-iCre-2A-eGFP.

### *In vivo* administration of rAAV vectors

Recombinant AAV vectors expressing luciferase driven by the CMV promoter or the CAG promoter were injected into the tail vein at a dose of $5 \times 10^{10}$ genomic particles (gp)/mouse under anesthesia with 2.5% isoflurane. Vectors expressing either iCre or Nemo were administered intravenously, and vectors expressing eGFP were administered intravenously, or—as alternative—intraperitoneally, intramuscularly, or subcutaneously (as indicated in the figure legends) at a dose of $1.8 \times 10^{11}$ gp/mouse in adults and $6 \times 10^{10}$ gp/mouse in neonates. Mice of individual experimental groups in each experiment were of similar age and weight and randomly allocated to vector treatment groups.

### Assessment of luciferase reporter gene expression

At different time points after vector injection, animals were anesthetized with 2.5% isoflurane. Luciferase expression was analyzed using a Xenogen IVIS200 imaging system (Caliper Lifesciences) and the software Living Image 4.0 (Caliper) after intraperitoneal injection of 200 μl luciferin substrate (150 mg/kg, Xenogen) per mouse.

Representative *in vivo* bioluminescence images were taken when luminescence in relative light units (photons/s/cm$^2$) reached the highest intensity. Three-dimensional reconstructions of *in vivo* luminescence images were obtained by using the DLIT option of the software Living Image 4.0 (Caliper) and measuring the emitted light in five different wavelengths from 560 to 640 nm, for 3 min each. The final figures showing bioluminescence images were assembled using the Photoshop (Adobe) software. After bioluminescence imaging, animals were sacrificed; the organs of interest were quickly removed, snap-frozen in liquid nitrogen, and stored at −80°C.

To quantify luciferase expression more accurately, organs were homogenized in reporter lysis buffer (RLB, Promega) using a Precellys 24 tissue homogenizer (Peqlab) with ceramic beads (CK28, Precellys, Peqlab). Luciferase reporter gene activity was determined in a luminometer (Mithras LB 940, Berthold Technologies) over a 10-s interval after adding 100 μl luciferase assay reagent (LAR, Promega) with a 2-s delay between each measurement. Values were normalized to total protein levels in each sample with the Roti Nanoquant Protein Assay (Roth).

**Analysis of vector biodistribution**

Fourteen days after i.v. administration of $5 \times 10^{10}$ genomic particles/mouse, total DNA from each organ was extracted using Precellys 24 tissue homogenizer (Peqlab) with ceramic beads (CK28, Precellys, Peqlab) and the DNeasy Tissue Kit (Qiagen) according to the manufacturer's instructions. DNA was quantified using a spectral photometer (NanoDrop ND-2000C (Peqlab)). AAV vector DNA in tissues was analyzed by quantitative real-time PCR using the CAG-specific primers 5′–GGACTCTGCACCATAACACAC–3′ and 5′–GTAGGAAAGTCCCATAAGGTCA–3 (see above). Vector copy numbers were normalized to total DNA.

**Transduction in cell culture**

Primary brain endothelial cells of mice (8 weeks of age) were cultured in 24-well plates as described previously (Ridder *et al*, 2011). Freshly purchased immortalized human brain endothelial cells (hCMEC/D3, free of mycoplasma, hepatitis A, B, C, HPV, herpes, and HIV 1 and 2; Millipore #SCC066) were grown in 24-well plates coated with rat tail collagen, type I (1:20 in 1× PBS) in EndoGRO-MV medium (Millipore # SCME004) supplemented with 1 ng/ml FGF2 (Millipore #GF003). Primary murine cells were infected with $1.0 \times 10^{10}$ gp/well, three days after preparation. Immortalized human cells were infected with $1.6 \times 10^{10}$ gp/well, one day after seeding. Medium was changed at least 3 days after infection. Ten days (murine cells) or 4 days (human cells) after infection, cells were fixed in 4% PFA and immunostained with chicken anti-GFP (1:2,000, Abcam #ab13970), and DAPI (1:2,000). Murine cells were additionally stained with rat anti-CD31, 1:500 (BD Pharmingen #557355). Infectivity was determined by the ratio of GFP-positive cells to DAPI.

**Immunohistochemistry**

Two weeks (in adults) or 8 days (in neonates) after vector injection, mice were anesthetized by isoflurane inhalation, followed by intracardial perfusion with PBS and 4% PFA fixation. Harvested organs

(brain, spinal cord, heart, kidney, muscle, liver, lung, and pancreas were postfixed in 4% PFA at 4°C for two hours, followed by one time wash in PBS and embedded in 2.5% agarose (in PBS)). Vibratome sections (60–100 μm) were prepared and stored in PBS at 4°C. Tissues for immunostaining were blocked in 5% BSA in PBS supplemented with 0.5% Triton X-100). Sections were incubated overnight at 4°C with the following primary antibodies: chicken anti-GFP, 1:2,000 (Abcam #ab13970); rat anti-CD31, 1:500 (BD Pharmingen #557355); rat anti-CD13, 1:400 (AbD Serotec #MCA2183GA); rabbit anti-aquaporin 4, 1:100 (Millipore #AB3594); rabbit anti-active caspase-3, 1:400 (Cell Signaling #9661S); or anti-collagen IV, 1:1,000 (Abcam #ab6586) in blocking buffer. The following day, sections were washed twice in PBS-Tx. Sections were later incubated in the following secondary antibodies for 2 h at RT: goat anti-chicken IgG, Alexa Fluor 488, 1:2,000 (Abcam #ab150169), anti-mouse-IgM-Cy3 1:400 (Jackson ImmunoResearch Laboratories), anti-mouse-IgG-HRP 1:5,000 (Santa Cruz Biotechnology, Inc.), or donkey anti-rat Cy3, 1:400 (Jackson ImmunoResearch #715-165-140). Sections were washed twice in PBS and mounted with Mowiol 4-88 containing 1,4-diazabicyclo-(2,2,2)octane.

**Microscopy**

Fluorescence images were taken at room temperature with one of the following microscopes:
Fluorescent microscope: DMI 6000 B (Leica); objectives: HCXPL FLUOTAR 10.0×, aperture 0.3, immersion; dry. HCXPL FLUOTAR L 20.0×, aperture 0.4, immersion; dry.
Confocal microscope: laser scanning microscope TCS SP5 (Leica); objectives: CPLAN 10.0×, aperture 0.4, immersion; dry. CPLAN L 20.0×, aperture 0.3PHI, immersion; water. HCXAPOLV-V-1 63×, aperture 0.9, immersion; oil.
Images were taken with a Leica DFC360FX camera using the acquisition software: LAS AF
Images were analyzed with ImageJ software. Images were not manipulated in any way except to make brightness and contrast adjustments. The final figures showing immunofluorescence images were assembled using the Illustrator (Adobe) software.

**Histological quantifications and determination of BBB permeability**

Endogenous immunoglobulin extravasation was quantified by using 20-μm cryosections after fixation in methanol and staining with Cy3-labeled donkey anti-mouse IgM antibody (see above). The integrated density of 2–4 images per mouse was determined by using the ImageJ software (National Institutes of Health). To quantify the string vessels, the lengths of collagen IV-positive and CD31-negative vessels were determined with ImageJ software. Total vessel lengths were measured in anti-CD31-stained sections. Two to four images were taken in the cortex and analyzed for all parameters. All cryosections from one experiment were always stained in parallel, and images were generated at the same settings.

Stainings for CD31 and collagen IV were quantified in a partially automated manner using a custom macro implemented into the image analysis software Fiji (www.fiji.sc/Fiji) (Schindelin *et al*, 2012). Briefly, images were thresholded using an auto-threshold method, despeckled, and smoothened to remove staining artifacts.

                                                                           

After converting the images to binary form, staining lengths were quantified with the "Skeleton 2D/3D" plug-in, yielding the number and lengths of stained structures in each image. To ensure that the quantification was reliable, the macro was initially compared to a manually quantified dataset showing a significant correlation ($r^2 = 0.9157$, data not shown). The BBB permeability was investigated as described previously (Ridder *et al*, 2015).

## Western blotting

The cerebella of mouse brains used in the string vessel analysis were homogenized in cell lysis buffer (Cell Signaling #9803) supplemented with PMSF (0.5 M) freshly prepared before use. Proteins were transferred to nitrocellulose membranes. The Western blots were probed overnight at 4°C with the following primary antibodies: goat anti-mouse IgG-HRP 1:2,500 (Santa Cruz # SC-2005), goat anti-mouse albumin 1:16,000 (Bethyl Laboratories #A90-134), and goat anti-actin 1:1,000 (Santa Cruz #sc-1615). Subsequently, HRP-conjugated secondary antibodies were added for 1–2 h at room temperature. For detection, we applied enhanced chemiluminescence (SuperSignal West Femto Substrate, Thermo Scientific) and a digital detection system (GelDoc 2000, Bio-Rad). Gel intensities were quantified with ImageJ software, and the ratio of albumin/actin or immunoglobulin/actin was measured.

## Animals

All mice were housed in individual, ventilated cages (IVCs) with 12-h light/dark cycles with food and water *ad libitum*. Experiments were performed under anesthesia with 2.5% isoflurane and 97.5% oxygen. Initial *in vivo* selection with the random AAV2-peptide library and analysis of AAV luciferase reporter vectors were performed in 8- to 12-week-old female FVB/N mice (Taketo *et al*, 1991). All mouse lines used for further experiments were established on a C57BL/6 background (Charles River). Mice were of both genders between the ages of 8–16 weeks, with the exception of animals in the $Nemo^{-/+}$ experiment, in which neonate animals were injected at P0. The Cre reporter line B6.Cg-Gt(ROSA) 26Sortm14(CAG-tdTomato)Hze/J (Ai14)(Madisen *et al*, 2010) was obtained from the Jackson Laboratory. Heterozygous *Nemo* knockout mice ($Nemo^{-/+}$) were generated by crossing $Nemo^{Fl}$ mice (Schmidt-Supprian *et al*, 2000) with *CMV-Cre* mice (Jax stock no. 6054). In experiments with $Nemo^{-/+}$ mice, five animals that did not show any characteristic skin phenotype were excluded from analysis. Only $Nemo^{-/+}$ mice with skin phenotype were analyzed at P8. Their body weights were measured over the 8-day period, and their brains were dissected either at P0 or at P8 for further analysis (see above). Brain endothelial-specific knockout ($Nemo^{beKO}$) animals were generated by crossing the BAC-transgenic $Slco1c1$-CreER$^{T2}$ strain (Ridder *et al*, 2011) that expresses the tamoxifen-inducible CreER$^{T2}$ recombinase under the control of the mouse $Slco1c1$ regulatory sequences with $Nemo^{Fl}$ mice (Schmidt-Supprian *et al*, 2000). $Nemo^{beKO}$ and $Nemo^{fl}$ mice were injected with AAV-BR1-CAG-NEMO or AAV-BR1-CAG-eGFP 7 days prior to tamoxifen treatment. To induce recombination, $Nemo^{beKO}$ and $Nemo^{fl}$ animals were injected i.p. with 1 mg tamoxifen dissolved in 90% Miglyol® 812/10% ethanol every 12 h for five consecutive days. Brains were dissected for analysis 7-10 days after the tamoxifen treatment. Littermates

### The paper explained

#### Problem
Efficient and tissue-specific gene therapy vectors are desperately needed for most clinically relevant targets. This is especially true for neurovascular diseases. Vectors based on adeno-associated virus (AAV) have been able to substantially improve the accessibility of the central nervous system (CNS). However, few of the existing vectors are tissue-specific after intravenous injection, and most CNS-directed AAV vectors preferably transduce neurons and/or astrocytes. Brain endothelial cells—as the key player of the blood–brain barrier—might display another promising site for gene therapy interventions which has not yet been targeted successfully in an efficient manner.

#### Results
By screening a random AAV display peptide library over five selection rounds in mice, we identified for the first time an efficient brain-homing AAV capsid mutant with a high degree of specificity for its target tissue. This mutant (designated AAV-BR1) mediated efficient and long-lasting transgene expression in BBB-associated endothelial cells after intravenous injection, whereas transgene expression in off-target organs was barely detectable. To demonstrate the therapeutic potential of AAV-BR1, we utilized this brain-specific capsid mutant to generate a gene therapy vector and were able to ameliorate the severe cerebrovascular pathology in a mouse model of incontinentia pigmenti.

#### Impact
This study demonstrates the high potential of screening random AAV display peptide libraries *in vivo* and highlights the importance of such libraries for the field of gene therapy. Since the applied technique is not limited to organs such as the brain, this study might help to develop vectors for a spectrum of other targets. This study further emphasizes the role of brain endothelial cells as relevant gene therapy intervention site and provides a potential therapeutic vector for a broad range of neurovascular diseases.

lacking the Cre transgene or age-matched wild-type animals were used as controls in all experiments. Investigators were blinded for treatment or genotype of mice, or both, in all experiments.

## Statistics

Statistical analysis was performed using the software GraphPad Prism 6 (GraphPad Software, San Diego, CA). All data were analyzed for normality by Kolmogorov–Smirnov test before applying parametric statistical tests. Datasets were analyzed either by Student's *t*-test or by one-way ANOVA (followed by Turkey's multiple comparison test), after being tested for differences in their variance by Brown–Forsythe test to ensure that groups of data with unequal sample sizes had similar variance, or data were tested by two-way ANOVA (followed by Bonferroni's multiple comparison test), as indicated in the figure legends. Based on data from previous projects or from preliminary experiments, we calculated the sample size using G*Power 3.1.9.2 to ensure adequate power of key experiments in detecting prespecified effect sizes. *P*-values < 0.05 were considered significant. The exact (Student's *t*-test) or adjusted (ANOVA) *P*-values for all experiments with indicated statistical significance are reported in the figure legends. Small datasets with samples sizes $n < 5$ were plotted as individual data points with bars representing the mean, without being tested for statistical significance when $n < 4$.

## Study approval

All animal experiments were performed in accordance with the European Community Council Directive of November 24, 1986 (86/609/EEC), the German Animal Welfare Act, and the ARRIVE guidelines. All efforts were made to minimize pain or discomfort of animals. Animal experiments were approved by the responsible local authorities and ethics review boards (Amt für Verbraucherschutz, Lebensmittelsicherheit und Veterinärwesen, Hamburg, Germany, and Ministerium für Energiewende, Landwirtschaft, Umwelt und Landwirtschaftsministerium ländliche Räume, Kiel, Schleswig-Holstein, Germany).

**Expanded View** for this article is available online.

## Acknowledgements

We are grateful to Jude Samulski, University of North Carolina, Chapel Hill, NC, for kindly providing the plasmids pXX2 and pXX6, to Rolf Sprengel, University of Heidelberg Germany for providing the plasmid p179, and to Beverly Davidson, University of Pennsylvania, Philadelphia, PA, for providing the plasmid pXX2-PPS. We thank Mascha Binder, University Medical Center Hamburg-Eppendorf, Germany, for helpful discussions and support. The luciferase expression of vector-treated animals was monitored with kind permission of and support by Axel Leingärtner at the UCCH Core Facility for Optical *in vivo* Imaging, University Medical Center Hamburg-Eppendorf, Germany. We also thank Beate Lembrich, and Julian Assmann, University of Luebeck, Germany, for expert support. This work was supported by the German Research Foundation (DFG, grants TR448/11-1 to MT and SCHW416/9-1 to MS) and the Margarethe Clemens Foundation (endowed professorship to MT).

## Author contributions

JK, GD, SM, MS, and MT designed the experiments. JK, GD, DAR, AH, JW, HS, ML, and JB performed the experiments. JK, GD, DAR, JAK, MS, and MT analyzed the data. MP contributed mouse strain. JK, GD, MS, and MT wrote the manuscript.

## Conflict of interest

The University Medical Center Hamburg-Eppendorf (UKE) has filed a patent application for the capsid-modified AAV vector BR1 on behalf of JK, SM, and MT. The authors declare that they have no additional conflict of interest.

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
