## [Review Process File · EMBO Molecular Medicine]

A BRAIN MICROVASCULATURE ENDOTHELIAL CELL-SPECIFIC VIRAL VECTOR WITH THE POTENTIAL TO TREAT NEUROVASCULAR AND NEUROLOGICAL DISEASES

Jakob Körbelin, Godwin Dogbevia, Stefan Michelfelder, Dirk A. Ridder, Agnes Hunger, Jan Wenzel, Henning Seismann, Melanie Lampe, Jacqueline Bannach, Manolis Pasparakis, Jürgen A. Kleinschmidt, Markus Schwaninger, Martin Trepel

Corresponding author: Martin Trepel, University Medical Center Hamburg-Eppendorf

Review timeline:

Submission date:	19 November 2015
Editorial Decision:	21 March 2016
Revision received:	28 February 2016
Editorial Decision:	15 March 2016
Revision received:	22 March 2016
Accepted:	23 March 2016

Transaction Report:

Editor: Roberto Buccione

1st Editorial Decision

21 March 2016

Thank you for the submission of your Report manuscript to EMBO Molecular Medicine. We have now received comments from the three Reviewers whom we asked to evaluate your manuscript

You will see that all three Reviewers are quite supportive of your work, although they do raise a few issues that prevent us from considering publication at this time. I will not dwell into much detail, as the evaluations are self-explanatory.

Reviewer 1 would like you to provide better immunocytochemical images, if available.

Reviewer 2 is more reserved and lists a number of important concerns for your consideration and careful discussion. Experimentally s/he would like further proof of tropism of the viral vector in a human setting that should not prove too laborious to perform.

Reviewer 3 also lists a few items to be addressed, including the suggestion to co-administer the targeting peptide with the vector as an added control.

The reviewers, and we agree, also ask for the provision of much more detailed description of experimentation, especially animal, including appropriate statistical treatment.

In conclusion, while publication of the paper cannot be considered at this stage, we would be

pleased to consider a suitably revised submission, provided that the Reviewers' concerns are addressed. As for further experimentation, the control experiment suggested by Reviewer 3 will not be a condition for acceptance, but I do suggest to incorporate additional data in this respect, should you have them available.

Please note that it is EMBO Molecular Medicine policy to allow a single round of revision only and that, therefore, acceptance or rejection of the manuscript will depend on the completeness of your responses, as outlined above, included in the next, final version of the manuscript.

As you know, EMBO Molecular Medicine has a "scooping protection" policy, whereby similar findings that are published by others during review or revision are not a criterion for rejection. Although I clearly do not foresee such an instance in this case, I do ask you to get in touch with us after three months if you have not completed your revision, to update us on the status. Please also contact us as soon as possible if similar work is published elsewhere.

Please note that EMBO Molecular Medicine now requires a complete author checklist (<http://embomolmed.embopress.org/authorguide#editorial3>) to be submitted with all revised manuscripts. Provision of the author checklist is mandatory at revision stage; The checklist is designed to enhance and standardize reporting of key information in research papers and to support reanalysis and repetition of experiments by the community. The list covers key information for figure panels and captions and focuses on statistics, the reporting of reagents, animal models and human subject-derived data, as well as guidance to optimise data accessibility.

I also suggest that you carefully adhere to our guidelines for publication in your next version, including presentation of statistical analyses and our new requirements for supplemental data (see also below) to speed up the pre-acceptance process.

I look forward to receiving your revised manuscript

***** Reviewer's comments *****

Referee #1 (Comments on Novelty/Model System):

This paper describes the generation and characterization of an AAV vector with high specificity for brain vascular endothelial cells following introduction into the systemic circulation. It comes at an opportune time, after reports that AAV2 can integrate into hepatic cells activating cancer driver genes (see: Nault et al., *Nature Genetics* 47, 1187-1193; 2015 and commentary, Russell and Grompe, same issue). Hence the authors should consider commenting in their discussion on potential safety advantages of this particular vector. Five rounds of selection resulted in the identification of a peptide, NRGTEWD showing the best homing potential for brain with minimal affinity for liver or heart. The efficacy and specificity of AAV expressing this capsid peptide (AAV-BR1) was compared to wild type AAV2 and a previously reported virus expressing the peptide DSPAHPS. The images presented leave little doubt as to the specificity of AAV-BR1. Expression in the eye is not unexpected, and perhaps the authors should comment upon the fact that both brain and inner retinal vascular endothelia are likely to be transfected. BR1 expression was monitored over prolonged periods and showed high stability in expression (Figure 3). These images are backed up by a quantitative assessment of luciferase activity in tissue lysates (Figure 4). Immunofluorescence studies, together with quantitative studies on transfected cells clearly indicated brain endothelial cells as the primary target for expression with some limited expression in neurons. The authors go on to use this vector in transfecting brain endothelial cells of Nemo \pm mice, having established an early window for therapeutic intervention. I found the immunocytochemical images a little unclear. The paper to which the Authors refer (Ridder et al, 2015) shows string vessels from a human normal and IP brain, which very clearly demonstrate what such string vessels look like. If the Authors have such available to include they should do so.

Referee #2 (Remarks):

Review of EMM-2015-06078

This article by K[^]rbelin and colleagues reports on a novel in vivo screening system of a broad library of random AAV capsid variants. The objective of this approach is to identify novel artificial AAV variants with high tropism for a specific tissue. In the context of this study, authors aimed at screening for an AAV viral vector with high CNS-targeting specificity and efficacy, and reduced off-target effects. Following intravenous injection of AAV2 variants containing random heptapeptide insertions in their capsid protein, brain tissues were harvested to amplify corresponding viral capsid DNA and to screen for an amino acid sequence motif that conveys brain-targeting properties. After a five round selection process, authors were able to narrow down their library to one promising AAV candidate (AAV-BR1), that shows highly improved and stable CNS transduction profile with almost no infectivity in peripheral tissues, when compared to both naturally occurring AAV2 and an artificial AAV2 variant generated for enhanced brain targeting, and which displays a phage-selected heptapeptide. The present study further demonstrates that this novel AAV-BR1 viral vector specifically infects brain vascular endothelial cells of the CNS. Therapeutic relevance of this new AAV-BR1 viral vector was then assessed on mouse models for incontinentia pigmenti (Nemo-/+ mice and conditional Nemo KO mice), a disease caused by loss-of-function mutations of the Nemo gene in brain endothelial cells, which leads to a disruption of the BBB. Authors show that intravenous administration of AAV-BR1-NEMO rescues endothelial cell survival and BBB functionality in these mice.

This is a thorough study, well executed and the manuscript is well written. In general, the results support the conclusions.

The following major comments should be addressed before publication:

-In this study authors have identified a new artificial AAV serotype with high tropism for brain endothelial cells. As indicated in the title, they claim that this viral vector might serve as a tool for future gene therapy approaches for neurovascular diseases. However, all experiments were performed either in vivo on mice or in vitro on primary brain endothelial cells prepared from mice as well. As the tropism of the viral vector might differ between the mouse and human species, it would be important to further validate this vector by including an experiment to demonstrate infectivity of human brain endothelial cells. This experiment would add great value to the study, and should be performed to support the conclusion that this newly discovered AAV2 variant is suitable for gene therapy against neurovascular diseases in humans.

-To assess the functional consequence of an AAV-BR1-Nemo-based gene therapy approach, authors used a conditional incontinentia pigmenti mouse model, where Nemo is deleted specifically in brain endothelial cells (Fig.8). As indicated in the Material and Methods' section of the manuscript, authors crossed Nemo floxed mice with a transgenic strain that expresses the tamoxifen-inducible CreERT2 recombinase under the control of the mouse Slco1c1 regulatory sequence. Conditional knock out of Nemo in brain endothelial cells is achieved by treating mice with tamoxifen. In this experiment, timing of tamoxifen treatment and intravenous administration of AAV-BR1-Nemo is crucial to fully appreciate the experimental procedures and therewith draw conclusions on the presented results. However, the Material and Methods section poorly describes the experimental procedures related to this part of the study, and more detailed instructions should be indicated.

-Overall, there is some confusion regarding the type of targeted neurological disorders. Is this novel vector suited to target "neurovascular diseases" (as stated in the title) or more broadly, "neurological disorders". The authors often allude to the possibility to use a vector targeting the CNS vasculature as a general tool for the treatment of CNS disorders (see for instance the first paragraphs of the Introduction and Discussion). It is unclear if the procedure of injecting AAV vectors to the brain parenchyma or CSF will be a major limitation to gene therapy for the CNS. Nevertheless, access to a vector with a specific targeting of the vascular endothelium in the CNS may indeed provide novel opportunities for the delivery of specific gene therapeutics to the CNS. Considering that mainly endothelial cells will be expressing the therapeutic factor, it remains however unclear to which type of neurological diseases this could be applicable. The authors should further discuss this important

question.

-In general authors should pay more attention to clearly indicate the age, at which mice received intravenous administration of the AAV-BR1 viral vector. Indeed, when reading through the Results' section, Figure legends, or Material and Methods' section of the manuscript, it is not specified which experiments were performed on neonatal mice and which ones on adult mice. It is only in the part of the Material and Methods' section describing the "In vivo administration of rAAV vectors" that the reader receives the information that AAV-BR1-Nemo has also been administered to adult mice, although it is unclear in which experiment this is the case. The authors should clearly indicate in the Results and/or Figure legends at which age the injections were performed in the animals and clarify the timeline of the experimental paradigm.

-Fig. 5 B, C & D: It would improve the manuscript if the authors could provide images at higher magnification to better demonstrate that the AAV-BR1-eGFP vector transduces the brain vascular endothelium. Furthermore, there is clearly a problem with the scale bars in these panels, since all three upper panels have the same scale bars but do clearly not represent the same magnifications. Authors need to correct this.

-Fig. 7 & 8: Whenever there are two independent variables (mouse genotype and treatment received) authors need to perform a two-way ANOVA instead of one-way ANOVA for statistical analysis.

Minor comments:

-Since expression profile of a given gene depends on AAV tropism as well as the promoter used, it is important to always associate these two indications for clear comprehension of the reader. The authors are therefore kindly suggested to indicate throughout their manuscript, which promoter was used for their study. For instance, at the page 7 of the manuscript, the authors write "Mice treated with AAV-BR1 eGFP vector...". We suggest authors to complete the vector description with "AAV-BR1-promoter eGFP..." for clarification.

Typographical mistakes:

-Page 9, line 4: "ref" in front of (Nenci et al, 2006) should be removed.

Referee #3 (Remarks):

The manuscript by Korbelen et al. describes an innovative in vivo screening system leveraging random library display using engineered adeno-associated viral capsids to select for ligand-directed vectors for tissue-specific gene delivery. The authors elected to study the brain to generate proof of concept data, considering the therapeutic relevance and specialized vasculature associated with the CNS. The library selection resulted in the identification of a viral capsid with highly favorable properties, namely, strong specificity in terms of tissue localization, and efficient gene transduction in the brain. The authors also demonstrated that the brain vascular endothelium is the primary target of transduction. Subsequently, they used a murine model of severe neurovascular pathology to evaluate the therapeutic effects of this ligand-directed gene delivery system. Administration of the targeted AAV vector that incorporated the defective gene relevant in this model completely reversed the vascular pathology that is characteristic of this disease.

General comments: This very important paper represents an exceptional conceptual advance, and is very likely to have a high impact in the field of therapeutic gene transfer. It addresses a very relevant and unmet clinical need that goes beyond AAV as a gene therapy vector. The implications are broad and might be applicable in many diseases, including, but not limited to the ones associated to the neurovascular system. The technology is not trivial in terms of reduction to practice, and for that, the team should be commended. The system is highly innovative, the data are convincing, and the

results are supported by impeccably performed experiments that reinforce the conclusions.

Major points:

1. The authors use i.v. injections as the route of vector administration, which is relevant vis a vis potential translational applications. It would be useful if the authors could elaborate on potential alternative routes of administration such as i.p. or s.c. injections.
2. It would be useful to evaluate whether or not co-administration of the targeting peptide affects the homing of the ligand-directed vector.

Minor points:

- AAV library production: the authors used the NNK principle for degenerated oligonucleotide synthesis (library insert). Other reports dealing with AAV libraries use NNB instead of NNK. The authors should justify why they designed their library this way.
- Vector biodistribution was studied 14 days after vector administration. Do the authors have any additional data on earlier time points? If so, these data should be included.
- The authors should discuss potential receptors in the brain vasculature as potential mediators of targeted localization.

1st Revision - authors' response

28 February 2016

Referee #1 (Comments on Novelty/Model System):

Comment:

1. *“This paper describes the generation and characterization of an AAV vector with high specificity for brain vascular endothelial cells following introduction into the systemic circulation. It comes at an opportune time, after reports that AAV2 can integrate into hepatic cells activating cancer driver genes (see: Nault et al., Nature Genetics 47, 1187-1193; 2015 and commentary, Russell and Grompe, same issue). Hence the authors should consider commenting in their discussion on potential safety advantages of this particular vector.”*

Reply:

We thank the reviewer for pointing out this very important safety aspect, which will certainly have implications on the field of AAV gene therapy. A comment on this point has been added to the discussion of the revised manuscript and the relevant references have been cited (*Discussion*: p. 10, end of first paragraph, lines 17-22)

Comment:

2. *“Five rounds of selection resulted in the identification of a peptide, NRGTEWD showing the best homing potential for brain with minimal affinity for liver or heart. The efficacy and specificity of AAV expressing this capsid peptide (AAV-BR1) was compared to wild type AAV2 and a previously reported virus expressing the peptide DSPAHPS. The images presented leave little doubt as to the specificity of AAV-BR1. Expression in the eye is not unexpected, and perhaps the authors should comment upon the fact that both brain and inner retinal vascular endothelia are likely to be transduced.”*

Reply:

We agree that the luminescence detected in the eye is most likely caused by vector-transduced endothelial cells of the retina and thank the reviewer for her/his advice. As the reviewer suggested, the detection of BR1-mediated transgene expression in the eye (which can be seen in Figure 2) is

now being mentioned in the results section of the revised manuscript and commented on in the discussion (*Results*: p. 6, lines 16 & 17. *Discussion*: p. 11, lines 24 & 25).

Comment:

3. “*BRI* expression was monitored over prolonged periods and showed high stability in expression (Figure 3). These images are backed up by a quantitative assessment of luciferase activity in tissue lysates (Figure 4). Immunofluorescence studies, together with quantitative studies on transfected cells clearly indicated brain endothelial cells as the primary target for expression with some limited expression in neurons. The authors go on to use this vector in transfecting brain endothelial cells of *Nemo*^{+/-} mice, having established an early window for therapeutic intervention. I found the immunocytochemical images a little unclear. The paper to which the Authors refer (Ridder et al, 2015) shows string vessels from a human normal and IP brain, which very clearly demonstrate what such string vessels look like. If the Authors have such available to include they should do so.”

Reply:

Due to the number of panels that are included in Figure 7 & 8, the size of the individual microphotographs is somewhat limited. To address the reviewer’s point and clearly demonstrate how string vessels look like, we enlarged the relevant panels in Figures 7 & 8 and included a more detailed microphotograph in the new Appendix, as Appendix Figure S4 (*Appendix*: p. 5. Referred to in the revised manuscript in *Results*: p. 8, lines 30 & 31).

Referee #2 (Remarks):

Review of EMM-2015-06078

*This article by Körbelin and colleagues reports on a novel in vivo screening system of a broad library of random AAV capsid variants. The objective of this approach is to identify novel artificial AAV variants with high tropism for a specific tissue. In the context of this study, authors aimed at screening for an AAV viral vector with high CNS-targeting specificity and efficacy, and reduced off-target effects. Following intravenous injection of AAV2 variants containing random heptapeptide insertions in their capsid protein, brain tissues were harvested to amplify corresponding viral capsid DNA and to screen for an amino acid sequence motif that conveys brain-targeting properties. After a five round selection process, authors were able to narrow down their library to one promising AAV candidate (AAV-BR1), that shows highly improved and stable CNS transduction profile with almost no infectivity in peripheral tissues, when compared to both naturally occurring AAV2 and an artificial AAV2 variant generated for enhanced brain targeting, and which displays a phage-selected heptapeptide. The present study further demonstrates that this novel AAV-BR1 viral vector specifically infects brain vascular endothelial cells of the CNS. Therapeutic relevance of this new AAV-BR1 viral vector was then assessed on mouse models for incontinentia pigmenti (*Nemo*^{-/+} mice and conditional *Nemo* KO mice), a disease caused by loss-of-function mutations of the *Nemo* gene in brain endothelial cells, which leads to a disruption of the BBB. Authors show that intravenous administration of AAV-BR1-NEMO rescues endothelial cell survival and BBB functionality in these mice.*

This is a thorough study, well executed and the manuscript is well written. In general, the results support the conclusions.

The following major comments should be addressed before publication:

Comment:

1. *“In this study authors have identified a new artificial AAV serotype with high tropism for brain endothelial cells. As indicated in the title, they claim that this viral vector might serve as a tool for future gene therapy approaches for neurovascular diseases. However, all experiments were performed either in vivo on mice or in vitro on primary brain endothelial cells prepared from mice as well. As the tropism of the viral vector might differ between the mouse and human species, it would be important to further validate this vector by including an experiment to demonstrate infectivity of human brain endothelial cells. This experiment would add great value to the study, and should be performed to support the conclusion that this newly discovered AAV2 variant is suitable for gene therapy against neurovascular diseases in humans.”*

Reply:

We agree with the reviewer that the specific tropism of the viral vector might differ between humans and mice. Unfortunately, viral infectivity might also differ between in vivo and in vitro experiments, due to potential changes in the expression pattern and the dedifferentiation of cells after taking them into culture (and / or immortalizing them) as it was shown for different populations of endothelial cells (Liaw & Schwartz, 1993; Staton et al, 2009). Additional differences between in vivo and in vitro studies comprise the interaction time for virus and receptor (short contact time in the blood flow vs. long lasting contact in cell culture medium), etc. These and additional reasons account for our technical approach of selecting the vector capsids upon the conditions they are eventually intended to be used in. Our published (Michelfelder et al, 2009) and unpublished work has revealed repeatedly, that vectors selected from AAV libraries on the target cells ex vivo do not function in vivo and vice versa. Therefore, ideally our vector's tropism towards human target cells would have to be checked in vivo, which is obviously not possible in humans at this point. As the reviewer suggested, we therefore evaluated (and confirmed) the infectivity of the BR1 vector on human brain endothelial cells (hCMEC/D3) which, like primary murine cells, were proven to be susceptible (Figure EV3). Thus, application of AAV-BR1 in the human setting might be possible, keeping in mind the limitations mentioned above. However, albeit certainly transducing human brain endothelial cells with substantial superiority to a previously published brain-endothelium-targeted AAV capsid harboring a control peptide, AAV-BR1 was equally infective as wild type AAV2 in the in vitro setting with hCMEC/D3 cells. This finding may either be explained by one of the reasons mentioned above or by inter-species differences (Lisowski et al, 2014). Despite having in mind the limited prognostic value of the in vitro experiment for the potential clinical benefit of AAV-BR1 in humans, we supplemented the revised manuscript with the new experimental data as Expanded View Figure EV3 and discussed them accordingly (*Results*: p. 8, first paragraph, lines 1-4. *Discussion*: p. 12, lines 12-21. *Materials & Methods*: p. 18, “Transduction in cell culture”. Figure EV3)

Comment:

2. *“To assess the functional consequence of an AAV-BR1-Nemo-based gene therapy approach, authors used a conditional incontinentia pigmenti mouse model, where Nemo is deleted specifically in brain endothelial cells (Fig.8). As indicated in the Material and Methods' section of the manuscript, authors crossed Nemo floxed mice with a transgenic strain that expresses the tamoxifen-inducible CreERT2 recombinase under the control of the mouse Slco1c1 regulatory sequence. Conditional knock out of Nemo in brain endothelial cells is achieved by treating mice with tamoxifen. In this experiment, timing of tamoxifen treatment and intravenous administration of AAV-BR1-Nemo is crucial to fully appreciate the experimental procedures and therewith draw conclusions on the presented results. However, the Material and Methods section poorly describes the experimental procedures related to this part of the study, and more detailed instructions should be indicated.”*

Reply:

We very much appreciate the thorough review of the experimental design of our study and thank the reviewer for this helpful advice. Missing important information of the experimental procedure of

this part of the study has been added to the revised manuscript (*Material and Methods*: p.21, “Animals”, lines 26-29)

Comment:

3. *“Overall, there is some confusion regarding the type of targeted neurological disorders. Is this novel vector suited to target “neurovascular diseases” (as stated in the title) or more broadly, “neurological disorders”. The authors often allude to the possibility to use a vector targeting the CNS vasculature as a general tool for the treatment of CNS disorders (see for instance the first paragraphs of the Introduction and Discussion). It is unclear if the procedure of injecting AAV vectors to the brain parenchyma or CSF will be a major limitation to gene therapy for the CNS. Nevertheless, access to a vector with a specific targeting of the vascular endothelium in the CNS may indeed provide novel opportunities for the delivery of specific gene therapeutics to the CNS. Considering that mainly endothelial cells will be expressing the therapeutic factor, it remains however unclear to which type of neurological diseases this could be applicable. The authors should further discuss this important question.”*

Reply:

Although the original title of our study stated “A ... gene therapy vector for NEUROVASCULAR diseases”, we believe that vectors like the ones described in our manuscript are very likely also applicable in non-vascular neurological disorders. It has been shown that neuronal lysosomal storage disorders can be treated in a mouse model by expressing the missing enzymes in brain endothelial cells, which subsequently secrete these enzymes into the brain parenchyma (Chen et al, 2009). Although this may open a broad range of neurological indications, we have confined this proof-of-principle evaluation with a primarily neurovascular disease, i.e., incontinentia pigmenti. To address the reviewer’s concerns and to avoid confusion, we discuss this question in the revised manuscript, as suggested (*Introduction*: p. 3, lines 10-14. *Discussion*: p. 10, lines 8-15) and we amended the title to “An adeno-associated viral vector with the potential to treat neurovascular and neurological diseases“.

Comment:

4. *“In general authors should pay more attention to clearly indicate the age, at which mice received intravenous administration of the AAV-BRI viral vector. Indeed, when reading through the Results' section, Figure legends, or Material and Methods' section of the manuscript, it is not specified which experiments were performed on neonatal mice and which ones on adult mice. It is only in the part of the Material and Methods' section describing the “In vivo administration of rAAV vectors” that the reader receives the information that AAV-BRI-Nemo has also been administered to adult mice, although it is unclear in which experiment this is the case. The authors should clearly indicate in the Results and/or Figure legends at which age the injections were performed in the animals and clarify the timeline of the experimental paradigm.”*

Reply:

We apologize for not specifying the age of mice clearly enough. All experiments, except those shown in Figure 7, were performed in adult animals. To make this more clear, information about the animals’ age has been added to all relevant figure legends and to the Results and the Material & Methods of the revised manuscript (*Results*: p. 9, lines 2, 16, 20. *Material & Methods*: p. 18, line 22. p. 19, line 6p. 21, lines 14 & 15. *Figure Legends*: p. 31-34, Expanded View Figure Legends p. 35). Again, we appreciate the reviewer’s thorough revision of the experimental design.

Comment:

5. *“Fig. 5 B, C & D: It would improve the manuscript if the authors could provide images at higher magnification to better demonstrate that the AAV-BRI-eGFP vector transduces the brain vascular endothelium. Furthermore, there is clearly a problem with the scale bars in these panels, since all three upper panels have the same scale bars but do clearly not represent the same magnifications. Authors need to correct this.”*

Reply:

The original version of the manuscript was submitted as one single document (including the figures). Apparently, the figure resolution was decreased upon conversion. This issue has now been fixed by submitting a high resolution figure together with the revised manuscript.

We apologize for mixing up the scale bars and thank the reviewer for advising us of this mistake. The high resolution version of Fig. 5 with corrected scale bars is now submitted as a new figure file (Fig.5. *Figure Legends*: p. 32).

Comment:

6. *“Fig. 7 & 8: Whenever there are two independent variables (mouse genotype and treatment received) authors need to perform a two-way ANOVA instead of one-way ANOVA for statistical analysis.”*

Reply:

Revised versions of Fig. 7 & 8 with proper statistical analysis have been submitted with the revised manuscript. The material and methods part and the figure legends (of all relevant figures) have been corrected regarding the statistics, including a more detailed description (*Material and Methods*: p. 22, “Statistics”. *Figure Legends*: p.32-34, Appendix Figure S3: *Appendix* p. 3).

Minor comments:**Comment:**

7. *“Since expression profile of a given gene depends on AAV tropism as well as the promoter used, it is important to always associate these two indications for clear comprehension of the reader. The authors are therefore kindly suggested to indicate throughout their manuscript, which promoter was used for their study. For instance, at the page 7 of the manuscript, the authors write “Mice treated with AAV-BRI eGFP vector...”. We suggest authors to complete the vector description with “AAV-BRI-promoter eGFP...” for clarification.”*

Reply:

We agree that indicating the utilized promoters is relevant. Where missing in the original, the utilized promoters have been indicated in the revised manuscript. (*Results*: p. 5, line 15. p. 7, lines 15, 16, 25. p. 8, line 7. P.9, lines 3, 4, 6, 8, 12, 10. *Figure Legends*: p. 31-34, *Expanded View Figure Legends*: p. 35)

Typographical mistakes:**Comment:**

8. *“Page 9, line 4: “ref” in front of (Nenci et al, 2006) should be removed.”*

Reply:

The mistake has been corrected.

Referee #3 (Remarks):

The manuscript by Korbelin et al. describes an innovative in vivo screening system leveraging random library display using engineered adeno-associated viral capsids to select for ligand-directed vectors for tissue-specific gene delivery. The authors elected to study the brain to generate proof of concept data, considering the therapeutic relevance and specialized vasculature associated with the CNS. The library selection resulted in the identification of a viral capsid with highly favorable properties, namely, strong specificity in terms of tissue localization, and efficient gene transduction in the brain. The authors also demonstrated that the brain vascular endothelium is the primary target of transduction. Subsequently, they used a murine model of severe neurovascular pathology to evaluate the therapeutic effects of this ligand-directed gene delivery system. Administration of the targeted AAV vector that incorporated the defective gene relevant in this model completely reversed the vascular pathology that is characteristic of this disease.

General comments:

“This very important paper represents an exceptional conceptual advance, and is very likely to have a high impact in the field of therapeutic gene transfer. It addresses a very relevant and unmet clinical need that goes beyond AAV as a gene therapy vector. The implications are broad and might be applicable in many diseases, including, but not limited to the ones associated to the neurovascular system. The technology is not trivial in terms of reduction to practice, and for that, the team should be commended. The system is highly innovative, the data are convincing, and the results are supported by impeccably performed experiments that reinforce the conclusions.”

Major points:**Comment:**

1. *“The authors use i.v. injections as the route of vector administration, which is relevant vis a vis potential translational applications. It would be useful if the authors could elaborate on potential alternative routes of administration such as i.p. or s.c. injections.”*

Reply:

Although we believe that intravenous injection will be the most favorable route of vector administration, we - like the reviewer- have also been interested in assessing alternative routes. Of note, all tested routes of administration (i.v., i.p., s.c., i.m.) are feasible ways to deliver the BR1 vector to the brain. However, i.v. injection proved to be most effective and most specific. We assembled the data about alternative injection routes into a new figure (Expanded view Figure EV5) and supplemented the revised manuscript by the additional data (*Results*: p.8, lines 13-17. *Material & Methods*: p. 17, lines 14-16.

Comment:

2. *“It would be useful to evaluate whether or not co-administration of the targeting peptide affects the homing of the ligand-directed vector.”*

Reply:

We thank the reviewer for raising this point. However, previous experience and theoretical considerations suggest that an effect of the soluble isolated peptide on vector homing is unlikely. In the past, we have performed blocking assays with other peptides that had been selected in the

context of an AAV library, without detecting any effect. The lack of blocking can be explained by the library system used for selection of the targeted vector. In our AAV display peptide libraries, unlike e.g. in phage display libraries, the peptides are embedded within the viral capsid which puts them into strong physicochemical constraints and, vice versa, may influence the structure of the surrounding capsid in a peptide sequence-dependent manner. This changes the biological behavior of the capsid profoundly. Therefore, our concept has not been to select for isolated peptides with defined properties but for whole viral particles that have been structurally changed in a certain manner by the insertion of such peptides. The situation in other systems like phage display is different. In the latter, the peptides are fused to one terminus of the phage's coat protein and are more or less freely accessible and under much less constraints than in an AAV capsid. This difference may explain why competitive blocking is possible with phage-selected peptides, whereas short peptides do not block gene transfer mediated by AAV library selected capsids. Thus, although we very much appreciate the reviewer's suggestion, we do not think that including the suggested experiment would add further conclusive information to the manuscript. These considerations can be included into the discussion of the paper, if reviewer and / or editor would like.

Minor points:

Comment:

3. *“2022; AAV library production: the authors used the NNK principle for degenerated oligonucleotide synthesis (library insert). Other reports dealing with AAV libraries use NNB instead of NNK. The authors should justify why they designed their library this way.”*

Reply:

Among the different codon schemes that are commonly used to generate random peptide libraries e.g. NNN, NNK, NNB (N = A/C/G/T, K = G/ T, B = C/G/T), both the NNK and the NNB scheme avoid two (UAA, AGA) out of three possible stop codons, which is an advantage compared to the NNN scheme because oligonucleotide inserts including a stop codon yield nonfunctional library clones. While the NNN scheme comprises all 64 codons and the NNB scheme 48 codons, the NNK scheme has the advantage to include only 32 codons. In all three schemes there are multiple codons for the individual 20 amino acids. A high redundancy of the nucleotide library leads to a low coverage of the peptide library, since more amino acids are encoded multiple times (for a discussion of the probabilistic assessment see (Nov, 2012)). Thus, the NNK scheme is clearly favorable over the NNB scheme in terms of a higher coverage (number of different peptide clones compared to the number of different oligonucleotide clones).

As a side note, a trimer-based library, with only one codon for each amino acid (no stop codons, highest coverage, equal distribution of all amino acids, etc.), would be even better. However, such trimer-based oligonucleotide libraries have not been affordable when we and other groups designed the AAV libraries some years ago. Although we agree with the reviewer that this is a very interesting topic, we feel that a detailed discussion might distract from the main message of the manuscript. Therefore, we only give a brief justification for using the NNK scheme in the Material & Methods section of the revised manuscript (*Material & Methods*: p. 14, “Preparation of the random AAV display peptide library”, lines 8 & 9.). If the reviewer and / or editor wish, this can be elaborated on in more detail in the final version of the manuscript.

Comment:

4. *“2022; Vector biodistribution was studied 14 days after vector administration. Do the authors have any additional data on earlier time points? If so, these data should be included.”*

Reply:

Unfortunately, we have not yet analyzed the biodistribution at an earlier time point than 14 days after vector administration. We agree that this question should be investigated in the future and thank the reviewer for the suggestion.

Comment:

5. “• The authors should discuss potential receptors in the brain vasculature as potential mediators of targeted localization..”

We agree that this is a very important question. As the reviewer suggested, we have included a discussion on potential candidates into the revised manuscript (*Discussion*: p. 11, end of page & 12, beginning of page)

Additionally to all the points mentioned above, we complemented the manuscript by mentioning a very recent study (Deverman et al, 2016) that might have implications for scientists working with AAV display peptide libraries (*Discussion*: p. 11, line 19-21). Deverman and colleagues describe as new cre-dependent selection system, which they used to identify AAV9 mutants with enhanced transduction of the CNS (neurons and astrocytes). However, albeit establishing this new selection methodology, the authors were not able to select for CNS specific AAV (and they left out brain endothelial cells as a key player of the BBB, the target of our AAV-BR1). Therefore, we are confident that the study by Deverman et al. does not impair the broad impact and novelty of the findings presented in our study.

References

- Chen YH, Chang M, Davidson BL (2009) Molecular signatures of disease brain endothelia provide new sites for CNS-directed enzyme therapy. *Nat Med* 15: 1215-1218**
- Deverman BE, Pravdo PL, Simpson BP, Kumar SR, Chan KY, Banerjee A, Wu WL, Yang B, Huber N, Pasca SP et al (2016) Cre-dependent selection yields AAV variants for widespread gene transfer to the adult brain. *Nat Biotechnol* 34: 204-209**
- Liaw L, Schwartz SM (1993) Comparison of gene expression in bovine aortic endothelium in vivo versus in vitro. Differences in growth regulatory molecules. *Arteriosclerosis and thrombosis : a journal of vascular biology / American Heart Association* 13: 985-993**
- Lisowski L, Dane AP, Chu K, Zhang Y, Cunningham SC, Wilson EM, Nygaard S, Grompe M, Alexander IE, Kay MA (2014) Selection and evaluation of clinically relevant AAV variants in a xenograft liver model. *Nature* 506: 382-386**
- Michelfelder S, Kohlschutter J, Skorupa A, Pfennings S, Muller O, Kleinschmidt JA, Trepel M (2009) Successful expansion but not complete restriction of tropism of adeno-associated virus by in vivo biopanning of random virus display Peptide libraries. *PLoS One* 4: e5122**
- Nov Y (2012) When second best is good enough: another probabilistic look at saturation mutagenesis. *Applied and environmental microbiology* 78: 258-262**
- Staton CA, Reed MW, Brown NJ (2009) A critical analysis of current in vitro and in vivo angiogenesis assays. *International journal of experimental pathology* 90: 195-221**

2nd Editorial Decision

15 March 2016

Thank you for the submission of your revised manuscript to EMBO Molecular Medicine. We have now received the enclosed reports from the referees that were asked to re-assess it. As you will see the reviewers are now globally supportive and I am pleased to inform you that we will be able to accept your manuscript pending the following final amendments:

1) I would suggest an alternative title explicitly mentioning vector specificity. Examples might be "AN ADENO-ASSOCIATED BRAIN MICROVASCULATURE ENDOTHELIAL CELL-SPECIFIC VIRAL VECTOR WITH THE POTENTIAL TO TREAT NEUROVASCULAR AND NEUROLOGICAL DISEASES" or "A BRAIN MICROVASCULATURE ENDOTHELIAL CELL-SPECIFIC VIRAL VECTOR FOR THE TREATMENT OF NEUROVASCULAR AND NEUROLOGICAL DISEASES" or variations of these.

2) As per our Author Guidelines, the description of all reported data that includes statistical testing must state the name of the statistical test used to generate error bars and P values, the number (n) of independent experiments underlying each data point (not replicate measures of one sample), and the actual P value for each test (not merely 'significant' or ' $P < 0.05$ ').

3) We are now encouraging the publication of source data, particularly for electrophoretic gels and blots, with the aim of making primary data more accessible and transparent to the reader. Would you be willing to provide a PDF file per figure that contains the original, uncropped and unprocessed scans of all or at least the key gels used in the manuscript? The PDF files should be labeled with the appropriate figure/panel number, and should have molecular weight markers; further annotation may be useful but is not essential. The PDF files will be published online with the article as supplementary "Source Data" files. If you have any questions regarding this just contact me.

Please submit your revised manuscript within two weeks.

I look forward to reading a new revised version of your manuscript as soon as possible.

***** Reviewer's comments *****

Referee #2 (Remarks):

The Authors have adequately revised the manuscript to address the comments of the Reviewer. This article is now suitable for publication.

Referee #3 (Comments on Novelty/Model System):

This manuscript represents a significant advance in the field, and it is well suited for the readership of EMBO Molecular Medicine.

Referee #3 (Remarks):

The authors have done an exceptional job addressing the referees comments and revising the manuscript accordingly.
I strongly recommend expedited publication.

2nd Revision - authors' response

22 March 2016

We are pleased to see that all three reviewers found their initial comments being adequately addressed and that they globally support publication of the revised manuscript.

You asked us to address three points before final acceptance of the manuscript.

1.) You suggest to change the title and to mention vector specificity. Combining the two proposals that you have given, we have made a "hybrid" of the two, as follows: "A BRAIN MICROVASCULATURE ENDOTHELIAL CELL-SPECIFIC VIRAL VECTOR WITH THE POTENTIAL TO TREAT NEUROVASCULAR AND NEUROLOGICAL DISEASES"

2.) You asked us to provide detailed description of the data that include statistical testing. We think that the relevant data are already described sufficiently in the revised manuscript. However, few minor clarifications have been made in the manuscript and the actual individual p-values which were listed as supplementary table have now been added to the individual figure legends (changes highlighted in yellow). Unfortunately, our statistical software (Graphpad Prism 6) does not calculate exact p-values if they are extremely small/ significant ($p < 0.0001$). Thus, exact numbers are only given for p-values ≥ 0.0001 . As minor amendment, the information n.s. (not significant) has been added to Fig.8f.

3.) The original Western blots for Figs. 7 & 8 will be provided as supplementary “Source Data”, as you suggested.

If there are any further open questions, please do not hesitate to contact us.

We thank you once more for your re-consideration of our manuscript and the positive feed-back.

We look forward to your reply.

Corresponding Author Name:

Journal Submitted to:

Manuscript Number: